# Aerosol dry deposition fluxes on snow during the ALPACA campaign in Fairbanks, Alaska

Antonio Donateo[1], Gianluca Pappaccogli[1,2], Federico Scoto[1,2], Maurizio Busetto[3], Francesca L. Lovisco[1,2], Natalie Brett[4,5], Douglas Keller[6], Brice Barret[7], Elsa Dieudonné[8], Roman Pohorsky[9], Andrea Baccarini[9,10], Slimane Bekki[4], Jean-Christophe Raut[4], Julia Schmale[9], Kathy S. Law[4], Steve R. Arnold[5], Gilberto J. Fochesatto[6], William R. Simpson[11], Stefano Decesari[3]

[1] Institute of Atmospheric Sciences and Climate (ISAC), National Research Council (CNR), Lecce, 73100, Italy
[2] Joint Research Center - ENI-CNR Aldo Pontremoli, Lecce, 73100, Italy
[3] Institute of Atmospheric Sciences and Climate (ISAC), National Research Council (CNR), Bologna, 40129, Italy
[4] LATMOS/IPSL, Sorbonne Université, UVSQ, CNRS, Paris, France
[5] Institute for Climate and Atmospheric Science, School of Earth & Environment, University of Leeds, UK
[6] Department of Atmospheric Sciences, College of Natural Science and Mathematics, University of Alaska Fairbanks, Fairbanks, AK 99775
[7] Laboratoire d'Aérologie, Université de Toulouse, CNRS, France
[8] Université du Littoral Côte d'Opale: Dunkerque, Hauts-de-France, FR
[9] Extreme Environments Research Laboratory, École Polytechnique Fédérale de Lausanne, EPFL Valais Wallis, Sion, Switzerland
[10] Laboratory of Atmospheric Processes and their Impacts, Ecole Polytechnique Fédérale de Lausanne, Switzerland
[11] Geophysical Institute and Department of Chemistry and Biochemistry, University of Alaska Fairbanks, Fairbanks, AK

*Correspondence to*: Antonio Donateo (a.donateo@isac.cnr.it) and Stefano Decesari (s.decesari@isac.cnr.it)

**Abstract.** A comprehensive study of aerosol exchange surface fluxes was conducted at a suburban site in Fairbanks (Alaska) during the Arctic winter as part of the ALPACA experiment. Aerosol fluxes were measured by an eddy covariance system on a snow-covered field located at the University of Alaska Fairbanks (UAF) Farm site from January 26th to February 17th, 2022. Overall, the flux measurements indicate that the site acted mainly as an emission source for ultrafine particles, while the fluxes for larger particle sizes were substantially bidirectional. Median deposition velocities were 0.61, 0.04, and 8.73 mm s$^{-1}$ for ultrafine (< 50 nm), accumulation (0.25 - 0.8 µm), and quasi-coarse (0.8 - 3 µm) particles, respectively. Anticyclonic synoptic meteorological conditions enhanced atmospheric stagnation and favoured pollutant accumulation near the surface, whereas cyclonic conditions increased aerosol dispersion, thus reducing deposition rates. Despite the frequent conditions of atmospheric stability and pronounced temperature inversions resulting from the strong surface radiative cooling, turbulence was generated mechanically by wind friction, leading to particle deposition. Our findings provide quantitative evidence that wintertime aerosol dry deposition in Arctic urban areas contributes significantly to pollutant accumulation in the snowpack, potentially enhancing contaminant remobilization during snowmelt. Finally, this study provides data for improving aerosol transport models and understanding pollutant-snow interactions in cold urban regions.

# 1 Introduction

The Arctic is a critical indicator of climate change, shaped by a complex interplay of physical, chemical, biological and socio-economic drivers and multiple feedback mechanisms with potential harmful impacts on environment and society. Among these drivers, air pollution significantly influences the Arctic climate, ecosystems, and public health (Prüss-Ustün et al., 2016; Schmale et al., 2021). Arctic air pollution encompasses harmful trace gases like tropospheric ozone, particulate matter (e.g. black carbon and sulphate), and other toxic substances (e.g. polycyclic aromatic hydrocarbons) (Law et al., 2014). Several studies on air pollution in the Arctic have been conducted over the years, with first reports about fine particulate matter and trace gases dating back to the 1950s (Mitchell, 1957): it was during this decade that the term "Arctic haze" was coined to describe the thick atmospheric aerosol layers whose anthropogenic nature was not yet understood at the time. In addition to long-range transport (Stohl, 2006), local emissions in developed Arctic regions significantly contribute to air pollution (Arnold et al., 2016). Socio-economic development and the consequent increase in urbanisation in certain areas of the Arctic can contribute therefore to degraded local air quality (Schmale et al., 2018). Specifically, local sources of air pollution across the Arctic regions include emissions from road transport (Weilenmann et al., 2009), residential heating employing wood, oil, coal, or natural gas, and electric power generation (AMAP, 2021). The combination of emissions with the unique winter weather conditions (characterized by e.g., a strong deficit in the surface energy budget) and the seasonal cryosphere dynamics, (Quinn et al., 2009; Willis et al., 2018) can exacerbate the exposure to air pollution and consequent human health impacts (Kovesi et al., 2007; Fuentes Leonarte et al., 2009). In the continental areas, winter conditions are characterized by radiative cooling and subsidence under high-pressure systems leading to persistent temperature inversions (Molders and Kramm, 2014) and inhibiting vertical mixing between the polluted air near the surface and the cleaner air above (Thomas et al., 2019a; Guo et al., 2020; Liu et al., 2021; Zhu et al., 2023). As a result, primary and secondary pollutants tend to accumulate within the urban atmospheric boundary layer (Mölders, and Kramm, 2010; Tran and Mölders, 2011; Molders et al., 2011). Extremely low temperatures further exacerbate air pollution by increasing per-capita energy consumption, a higher number of heating-degree days, elevated emissions from cold-engine starts, and frequent short-distance driving. Further, the aerosol scavenging process in ice precipitation is less efficient than in warm clouds leading to enhanced transport and atmospheric lifetime of particulate pollutants (Arnold et al., 2016). Despite these significant differences in chemistry and meteorological conditions compared to cities at mid- or low latitudes, Arctic urban air quality remains a relatively underexplored area, with only a limited number of intensive field studies addressing this issue (Mölders and Kramm, 2018).

Snow is ubiquitous in many important polar and subpolar urban environments. The significance of snow and ice in regulating regional climate, aerosol-cloud interactions, atmospheric chemistry and biogeochemical cycles has been well established in remote regions (Thomas et al., 2019b). However, the role of snow and ice physics and chemistry in urban areas - where numerous air-pollutant emission sources are concentrated - remains poorly understood (Ariya et al., 2018). Snow, with its porous structure and extensive winter coverage, effectively scavenges contaminants, also enabling significant exchange of trace gases between the surface of the snowpack and the atmosphere (Grannas et al., 2007). During the long-lasting anticyclonic

conditions in winter (Reeves and Stensrud, 2009; Largeron and Staquet, 2016), when the shallow thermal inversions trap anthropogenic pollutants near the surface (Simpson et al., 2024; Brett et al., 2025; Pohorsky et al., 2025), atmosphere-cryosphere interactions can take place through aerosol deposition and chemical transformations of organic and inorganic components in the snowpack. These processes were shown to potentially impact urban air quality, especially in areas characterized by a complex terrain (Kuoppamäki et al., 2014; Osipova et al., 2015; Nazarenko et al., 2017; Wei et al., 2017; VanReken et al., 2017).

Fairbanks (Alaska, USA), with a population of approximately 33,000 is the second largest city and one of the most polluted cities in Alaska. Numerous studies have investigated the causes of high pollution levels and sources of particulate matter in the city (Tran and Mölders, 2011; Mölders and Kramm, 2018; Robinson et al., 2022, 2023). Observations, combined with photochemical modelling, show that the region receives only minor amounts of pollution from long–range transport (Cahill, 2003; Tran et al., 2011). The major sources of primary particulate matter are strong emissions from local sources, in combination with a poor dispersion of pollution that occurs during the winter months with extreme cold conditions (Robinson et al., 2023). During winter, observed daily mean concentrations often exceeded the United States (US) 24-h National Ambient Air Quality Standard (NAAQS) of 35 µg m$^{-3}$, particularly under conditions of calm winds, extremely low temperatures (< -20°C) and low moisture (water-vapor pressure < 2 hPa) during prolonged inversions (Tran and Mölders, 2011).  In the framework of the ALPACA (Alaskan Layered Pollution And Chemical Analysis) project, a multi-disciplinary observational campaign was carried in winter 2022 in Fairbanks to investigate the sources of air pollution, pollutants transformations, and meteorological conditions contributing to urban air quality related issues (Simpson et al., 2024).

The deposition of particles containing toxic metals, pesticides, polyfluorinated compounds, or persistent organic pollutants in an urban context is an emerging topic of concern (Hageman et al., 2010; Casal et al., 2017; Farmer et al., 2021). Dry deposition is a complex process that is influenced by the microphysical properties of aerosols and their sources, meteorological conditions, and surface morphological characteristics (Donateo and Contini, 2014; Mohan, 2016; Farmer et al., 2021; Donateo et al., 2023). The number of representative datasets for the Arctic urban areas is limited with few cases of aerosol deposition measurements on snow surfaces in an urban or suburban area (Duann et al., 1988). Urban surfaces present a particularly challenging environment to study, due to the complexity of airflows and micrometeorology in these areas.  This makes our work one of the few attempts to investigate deposition processes under such conditions. Eddy covariance (EC) measurements are challenging and rarely carried out, although several recent studies have successfully characterized urban emissions of particles essentially from vehicle exhaust and other sources (Märtensson et al., 2006; Jarvi et al., 2009; Contini et al., 2010; Deventer et al., 2018; Donateo et al., 2019). At the same time, it is important to acknowledge that EC measurements in these environments may be subject to additional uncertainties, for example due to low turbulence, surface heterogeneity, or snow-related effects on particle exchange. The atmospheric burden of aerosol compounds and its change between the present day and preindustrial conditions are sensitive to the representation of dry deposition processes in the global climate models (Clifton et al., 2024). Atmospheric aerosol dry deposition is very uncertain and often parameterized based on sparse field observations (Nilsson and Rannik, 2001; Saylor et al., 2019). The rapid environmental change observed in the Arctic over the recent period

highlights the critical need to improve our understanding of the processes driving the sources, transport, and impacts of Arctic air pollutants, as well as their effects on Arctic communities. However, limited predictive capabilities and a lack of observations in high-latitude regions present major obstacles to advancing this understanding and to producing reliable short- and long-term projections of Arctic environmental changes (Arnold et al., 2016; AMAP, 2021). The primary aim of the present work is to investigate the meteorological processes affecting particle deposition in a polluted Arctic urban environment. The study focuses on characterizing aerosols, measuring size-segregated particle fluxes (from ultrafine to quasi-coarse), and determining their dry deposition rates. The findings of this study provide valuable insights into dry deposition processes on urban snow cover and help refine predictive models for dry deposition in sub-Arctic urban environments. Section 2 describes the methodology with details on the eddy covariance instrumental details as well as data processing of micrometeorological data. Site meteorology and dynamic processes influencing the surface boundary layer are described in Sect. 3.1. The observed size-segregated particle concentration and exchange fluxes are discussed in Sect. 3.2-3.3. The analysis of the particle deposition velocity and its relationship with the meteorological conditions is shown in Sect. 3.4-3.6. Finally, a discussion of the relationship between deposition phenomena and boundary layer vertical structure is presented in Sect. 3.7.

## 2 Methods

### 2.1 Measurement Site

Aerosol fluxes were measured at the University of Alaska Fairbanks (UAF) Farm site (Fig.1) located in the northwest outskirts of Fairbanks (64° 51' 12.8" N, 147° 51' 34.7" W) by an eddy covariance (EC) system. The measurement campaign started on 26th January 2022 and lasted until 17th February, for a total of 23 days. The measurement site was located at the foothills of the mountainous terrain enclosing the Tanana River basin from north and at the mouth of the Goldstream valley, which is a small tributary of the Tanana valley. The Fairbanks area, along the Tanana River, is predominantly flat, even if a series of hills are present, particularly to the west and north sector. The ground is generally frozen for most of the year, with ubiquitous snow cover in winter. The Fairbanks International Airport is located about 6 kilometres west of downtown and 4 kilometres south-east of the UAF farm. In the Fairbanks area, five power plants are present, and they may contribute to surface pollution through emissions of trace gases and particulate matter (Brett et al., 2025). Meanwhile, residential heating and transportation represent the primary anthropogenic sources of these pollutants, significantly impacting air quality (Ijaz et al., 2024). A more detailed map of local sources, including the airport and power plants, can be found in Brett et al. (2025, Fig. 1).

### 2.2 Instrumental set-up

The measurement system was deployed on the rooftop of a container at 11 m above ground level, mounted on a pneumatic mast. The eddy covariance (EC) station consisted of an ultrasonic anemometer (Gill R3, Gill Instruments Ltd., Lymington, UK) operating at 100 Hz, a condensation particle counter (CPC, TSI 3756) providing total particle number concentrations at 1 Hz, and an optical particle counter (OPC, Grimm 11-D) resolving particle number concentrations across 16 size bins ranging

from 0.25 to 3 µm, also at 1 Hz. Further details regarding the instrumentation, inlet tubing, and system configuration are provided in Donateo et al. (2023). The aerosol sampling system operated at a nominal flow rate of 60 L min⁻¹, under turbulent conditions characterized by a Reynolds number of 4371, with continuous flow monitoring via a digital flowmeter (TSI, model 4043). The CPC was connected to the flow splitter through a 0.17 m-long conductive silicon tube (6 mm internal diameter), sampling at 1.5 L min⁻¹, while the OPC sampled through a 0.30 m-long tube (4 mm internal diameter) at 1.2 L min⁻¹. Average total particle losses in the sampling lines were 12% for the CPC and approximately 0.3% for the OPC. Penetration curve analysis (Kupc et al., 2013) indicated a 50% cutoff diameter ($d_{50}$) at around 5 nm for the CPC.

Meteorological variables were concurrently recorded at the top of the mast using a conventional thermo-hygrometer (Rotronic XD33A). Complementary measurements of air temperature, humidity, wind speed, and radiation were performed above the snow surface using a meteorological station equipped with a HygroVUE10 sensor (Campbell Scientific, UK), a heavy-duty wind monitor (R. M. Young, USA), and a four-component radiometer (SN-500, Apogee Instruments Inc., USA) (Pohorosky et al., 2025).

**2.3 Eddy covariance data analysis**

Atmosphere–surface turbulent fluxes of aerosol particles were quantified by applying the eddy covariance method to 30-minute averaging periods. By combining CPC and OPC measurements, a broad aerosol size spectrum was characterized, spanning ultrafine to quasi-coarse particles across 17 size bins (geometric mean diameters: 0.035, 0.26, 0.29, 0.32, 0.37, 0.42, 0.47, 0.54, 0.61, 0.67, 0.75, 0.89, 1.14, 1.44, 1.79, 2.24, and 2.74 µm). For each particle size bin (index $i$), particle fluxes from particle number concentration $N_i$ were calculated according to $F_{Ni} = \overline{w'N_i'}$ [cm⁻² s⁻¹] where $w$ represents the vertical wind velocity. Size-resolved exchange velocities $V_{ex}$ [mm s⁻¹] were defined as the normalized turbulent fluxes:

$$V_{exi} = -\frac{F_{Ni}}{N_i} \qquad (1)$$

namely the turbulent flux of each stage normalized by the respective particle number concentration. A negative particle flux corresponds to a positive exchange velocity (hereafter $V_d$), indicating transport toward the surface (deposition), whereas a positive particle flux corresponds to a negative exchange velocity ($V_e$), representing transport into the atmosphere (emission). Alongside particle fluxes, key turbulence parameters were derived, including virtual sensible heat flux $H = \rho\, c_p \overline{w'T'}$ (hereafter referred to as sensible heat flux), where T denotes the sonic temperature, $c_p$=1005 J kg⁻¹ K⁻¹ is the specific heat at constant pressure, and ρ represents air density. Additionally, the turbulent kinetic energy $TKE = \frac{1}{2}(\sigma_u^2 + \sigma_v^2 + \sigma_w^2)$, where $\sigma_u$, $\sigma_v$, $\sigma_w$ are the standard deviations of the wind velocity components. Lastly, the friction velocity is defined as $u^* = (\overline{u'w'} + \overline{v'w'})^{1/4}$. Atmospheric stability was characterized by the dimensionless parameter $\zeta$ = z/L, where z is the measurement height (11 m) and L is the Obukhov length (Stull, 1988). The measurement campaign was dominated by stable conditions ($\zeta$ > 0.01) in 46% of cases, with very stable stratification ($\zeta$ > 1) in 30%, and unstable ($\zeta$ < −0.01) and very

unstable ($\zeta$ < −1) conditions observed less frequently (12% and 11%, respectively). Neutral conditions were rare (< 1%) (Nordbo et al., 2013).

Data were flagged for discontinuities caused by power loss, or values outside the absolute limits, and discarded from the dataset. The total data coverage during this experiment was 71% for the anemometer, 76% for the CPC and 83% for the OPC, respectively. The raw data (100 Hz) were pre-processed applying a despiking procedure and a replacement data by linear interpolation (Mauder et al., 2013). The EC fluxes measured by a closed path instrument (i.e., CPC or OPC) need to be corrected for the time delay (time lag) between the vertical wind component fluctuations and the particle concentration fluctuations. Time lag was determined by a cross-correlation analysis (Deventer et al., 2015), yielding average lags of 5.38 s for the CPC and 5.30 s for the OPC (for all size channels), respectively. To minimise the anemometer tilt error, a three-dimensional coordinate system transformation was applied to the data set, using the planar fit method proposed by Wilczak et al. (2001). The planar fit coefficients are calculated for the whole campaign period. The fit coefficients were calculated over the whole direction around the pneumatic mast. In neutral or very stable atmospheric conditions with low wind speed, weak and intermittent turbulence, the sub-meso motions do not follow surface-layer similarity (Sun et al., 2012, Schiavon et al., 2019). In this work, the energy contributions related to sub-meso motions and instrument drifts were removed by a recursive digital filter both for energy and particle fluxes (Falocchi et al., 2018; Pappaccogli et al., 2022, Donateo et al., 2023). Further, the low-frequency loss due to finite averaging time and filtering procedure was corrected following Burba et al. (2022). Stationarity was assessed following Mahrt (1998), and non-stationary data (16% for ultrafine, 9% for accumulation, and 3% for quasi-coarse modes) were excluded from further analysis. A lower detection limit for the fluxes in the sampling system was computed using the method proposed by Langford et al. (2015) as 2.8 cm$^{-2}$ s$^{-1}$ for the CPC and 0.2 cm$^{-2}$ s$^{-1}$ for the OPC. Error associated with the random and limited statistical counting (relative error, %) was estimated through the approach reported in Deventer et al. (2015) for particle number concentration $\delta(N)$ and fluxes $\delta(F_N)$. The method reported in Fairall (1984) was used for the deposition velocity $\delta(V_d)$ for each size range. If the counting errors on deposition velocity $\delta(V_d)$ are considered, on the first size channel (CPC) it was very low (< 1%). The same error for the first eleven channels of OPC (0.25 μm - 0.80 μm) was on average 64%, while for the remaining channels (1 μm - 3 μm) it was on average 101%. To lower the associated statistical counting error, especially on deposition velocity, the first nine channels of the OPC have been pooled together as have the rest of the seven channels (Whitehead et al., 2012; Donateo et al., 2023). To assess the independence of the particle size modes, we calculated Pearson correlation coefficients between number concentrations in adjacent OPC size channels, merging those with a correlation > 0.5. This approach reduces coverability between size classes and ensures that the reported modes represent distinct particle populations for interpretation of sources and deposition processes. Based on this aggregation, particles concentration are segregated into three size ranges according to the particle diameter ($d_p$): the ultrafine (UFP, 5 nm < $d_p$ < 0.25 μm), the accumulation (ACC, 0.25 < $d_p$ < 0.7 μm), and the quasi-coarse (Q-CRS, 0.8 < $d_p$ < 3 μm) mode, the last indicating a size range between large accumulation mode and small coarse particles. Ultrafine particle concentration (UFP) was obtained as the difference between the total number concentration (CPC measurement) and the OPC

integrated concentration in the size range 0.25 - 1 μm. The relative counting errors on $V_d$ are 16% for ACC and 41% for Q-CRS. The first-order time constant of the CPC and the OPC measurement systems was determined by estimating the time response (at first order) to a concentration step with the campaign setup configuration. The results were $\tau_{CPC} = 0.6 \pm 0.2$ s and $\tau_{OPC} = 0.23 \pm 0.06$ s (identical for each size channel). High frequency losses were corrected following the parametric/in situ approach developed by Horst (1997) and they have been quantified on average in 23% for the CPC and 12% for the OPC.

The aerodynamic roughness length ($z_0$) was estimated under neutral conditions as 0.006 m, consistent with typical values for snow-covered surfaces (Weill et al., 2012; Helgason and Pomeroy, 2012; Maillard et al., 2024). No significant statistical differences were found exploring the roughness length for different wind sectors. In the UAF Farm site a null displacement height has been considered, not being significant aerodynamic obstacles. Source area for scalar fluxes has been evaluated using the footprint model proposed by Kljun et al. (2015). Results of flux footprint analysis of the EC system are shown in Fig. 1. The EC footprint stretched in the west, north-west sector over an open space, marginally intercepting Parks Highway for about 1.1 km (Fig. 1). The footprint extended about 400 m to the north-east, and about 240 m to the south-west and south-east direction. The flux peak contribution was in the W-NW sector at about 80 m (Fig. 1). Except for the presence of the road, the area was essentially lacking anthropogenic emissions, and the ground was homogeneous in all wind direction sectors around the EC site, with full snow coverage during the measurement period.

## 3 Results and Discussion

### 3.1 Site Meteorology

The meteorological situation at the UAF Farm site during the ALPACA measurement campaign was presented in previous works (Simpson et al., 2024; Brett et al., 2025; Pohorosky et al., 2025). The meteorological conditions during the campaign were characterized by alternating anticyclonic (AC) and cyclonic (C) periods, with two distinct transition phases (T1 and T2), as described in detail by Pohorosky et al. (2025). An anticyclonic period (AC) considered in the present study lasted from January 25[th] to February 1[st]. This was followed by a transition period (T1) from February 2[nd] to 3[rd], during which a low-pressure system moved north-eastward from the Aleutians, creating a north-south high-low pressure gradient over Fairbanks. Starting on February 4[th], a series of secondary lows formed off the main Aleutian low and moved northward, maintaining cyclonic conditions over Fairbanks until February 10[th] (C period). In the days that followed, a persistent Siberian high-pressure system intermittently extended and connected with a high-pressure system over the Gulf of Alaska, marking the second transition period (T2) from February 11[th] to 18[th] (Fig. 2).

Air temperature was below 0 °C for the whole measurement campaign (Fig. 2a). It was lower during the AC days with an average of -27.3 °C with respect to the cyclonic period (-20.3 °C). The minimum temperature (-34.4 °C) was reached on 30[th] January in the morning. The temperature difference between the top of the EC system (11 m a.g.l.) and the ground-level sensor (2 m a.g.l.) can be used as a measurement of the surface-based inversion (SBI) strength (Fig. 2a) (hereafter ΔT). The anticyclonic period was characterised by frequent (about 6 events in 10 days), long-lasting intense SBIs. On average, ΔT in

the AC period was 2.4 °C (with a mean T gradient of 0.3 °C m$^{-1}$), with a maximum value of 13.9 °C during the night of 3$^{rd}$ February and extending for more than 2 days. During the calm and clear-sky nights typical of anticyclonic phases, surface radiative cooling develops into negative temperature vertical profiles progressively as the hours pass, hence with stronger gradients observable especially in the late night and early morning hours before dawn. Moreover, during winter anticyclonic periods, inversions can persist for several days, as the lack of wind and turbulence inhibit air mixing, with their strength somewhat modulated by the (weak) solar irradiation in the middle of the day (Fig. 2a). During the cyclonic period, the intensity of the temperature inversion ΔT was reduced with a mean value of 0.7 °C and, in general, less frequent, often not detectable by the temperature difference between 2 and 11 m at the EC site, although inversions could still be detected at higher elevations during some of the balloon operations in T and C periods but in any case less strong than during the AC period (Pohorsky et al., 2025). At a local scale, at the UAF Farm site, two prevailing wind directions can be distinguished: one characterised by winds coming from the north-west for 69% of the cases and one with winds coming from the south-east for 16% of the cases (Fig. 2b). There was also a wind component in the north-east sector for 7% of cases. The highest wind speeds (on average 2.85 m s$^{-1}$) were measured from the north-west direction with a maximum wind speed of 6.1 m s$^{-1}$, while in the south-east direction (on average 1.18 m s$^{-1}$) the maximum wind speed was about 2.46 m s$^{-1}$. During the ALPACA campaign, the prevalent wind circulation at UAF Farm at high wind speeds from NW was characterised by a katabatic flow or "Shallow Cold Flow" (SCF) from the mouth of the Goldstream valley into the Tanana basin which sometimes reached wind speeds of 5 m s$^{-1}$ at surface level (Fochesatto et al., 2015). In Maillard et al. (2022), wind lidar measurements showed that the north-westerly SCFs were contrasted by north-easterlies above ca. 80 m a.g.l. Also, in absence of SCFs, a weak westerly flow generally prevailed at UAF Farm. During daytime, a slow reverse flow from the southeast often emerged, bringing urban pollution from downtown Fairbanks.

The sensible heat flux was negative for most cases (76%) with a mean value of -3.95 W m$^{-2}$ (median -1.93 W m$^{-2}$) over the whole campaign (Fig. 2c). The sensible heat flux reached its minimum value (maximum as absolute value) during the anticyclonic period (-70.42 W m$^{-2}$) when the average value was -6.12 W m$^{-2}$. By contrast, during the cyclonic period, H reached a minimum value (maximum as absolute value) of -21.85 W m$^{-2}$ with a mean of only -1.86 W m$^{-2}$. As the surface was colder than the atmosphere above for most of the time, H tended to be negative (i.e., the air warms the surface). The potential for negative sensible heat flux became more prominent when the atmospheric temperature vertical profile was steeper and the surface coldest. These conditions were prevalent during the anticyclonic period of the campaign, where they were interrupted by surface-heating processes due to downwelling longwave radiation (on cloudy days such as 27$^{th}$ Jan) or shortwave (solar) radiation (on clear days, especially approaching the end of the campaign). It is worth noting that during the AC period, there was a daily-scale alternation between long hours (in the dark) of very negative net surface radiation with NW winds (often accompanied by high wind speeds, stronger than 3 m s$^{-1}$) and short daytime periods of less negative net radiation fluxes often associated with weak SE flows. High sensible heat flux values were often found in the AC period. By contrast, in the following T1 and C periods, only moderately low net surface radiation fluxes were found with a null daily variability (because of the cloudy sky), generally low winds speeds and small sensible heat flux values. The appearance of two meteorological regimes

characterized by different wind speed and net surface radiation conditions at this site is described by Maillard et al. (2022) and can be put in relation to the shape of the temperature profile in the lower atmosphere as reported by Pohorsky et al. (2025). It was found that, even if stronger thermal inversions were normally found in the AC period, the wind stress promoted the formation of a surface mixing layer lowering the temperature gradient near the ground and leaving a layer above characterized by a very steep temperature gradient. By contrast, during the T and C periods, low surface winds and lesser net surface cooling were responsible for a homogeneous temperature gradient from the surface to the height of inversion (typically at ca. 50 - 60 m a.g.l.). The implications of these boundary layer features for surface fluxes will be described in the sections below.

## 3.2 Particle Concentration

Average particle number concentration N, over the whole measurement period, for UFP (hereafter $N_{UFP}$) was 16,849 cm$^{-3}$ (median 12,767 cm$^{-3}$), for ACC mode (hereafter $N_{ACC}$) was 92 cm$^{-3}$ (median 76 cm$^{-3}$), and for Q-CRS mode (hereafter $N_{Q-CRS}$) was 0.35 cm$^{-3}$ (median 0.29 cm$^{-3}$) (Table 1). $N_{UFP}$ shows very high concentration for the first two days (26$^{th}$ and 27$^{th}$ January) of the measurement campaign with an average amount of 42,831 cm$^{-3}$. Also, $N_{ACC}$ presented a very intense peak in the concentration values (162 cm$^{-3}$) in these two days, while $N_{Q-CRS}$ is not very different from the whole average (0.42 cm$^{-3}$). The particle number concentrations observed in this study are consistent with previous measurements reported for Fairbanks during the winter season. For example, Robinson et al. (2023) documented a median particle number concentration above $4.5 \times 10^4$ cm$^{-3}$ during cold stagnation events, with UFPs accounting for most particles (> 95%). Again, Robinson et al. (2023), measured the highest UFP number concentration (7.2 x 10$^4$ cm$^{-3}$) in Downtown East (Fairbanks). The particle number concentrations observed in this study in the immediate outskirts of Fairbanks are comparable to, or slightly higher than, those previously reported in the surroundings of Fairbanks. For instance, Robinson et al. (2023) measured concentrations on the order of 1.5 x 10$^4$ cm$^{-3}$ at sites located on the hills north of the city during strong inversion conditions. By contrast, typical particle number concentrations at pristine Arctic sites, such as Barrow in Alaska (Rose et al., 2021) or Zeppelin observatory in Ny-Ålesund (Croft et al. 2016) are two to three orders of magnitude lower (10$^2$ - 10$^3$ cm$^{-3}$), underscoring the dominant impact of local sources and boundary-layer processes in shaping aerosol levels in Fairbanks. Our observations thus align with pollution episodes previously described for the region and highlight the strong contrast between clean background conditions and the highly elevated concentrations associated with persistent inversions and limited boundary-layer mixing.

After one day characterized by a synoptical-scale advection and boundary layer ventilation (28$^{th}$ January), a "cold pollution event" developed from January 29$^{th}$ until the early afternoon of February 3$^{rd}$ (Simpson et al., 2024). This event exhibited the coldest conditions of the study period, with temperatures of −20 to −38 °C (Fig. 2). From January 29$^{th}$ to February 1$^{st}$, it was observed an increase in the aerosol content in all the size ranges, from UFP to Q-CRS mode (Fig. 3). This period was dominated by an anticyclone promoting air mass stagnation, with surface wind speeds of less than 3 m s$^{-1}$ (2.63 m s$^{-1}$ on average), and a local circulation driven by sub-mesoscale flows (e.g., from valley-ridge thermal gradients, Fochesatto et al., 2015). Under these conditions, atmospheric stagnation enhanced pollution levels and the accumulation of aerosols in all size ranges (UFP, ACC and Q-CRS) (see Table 4 and Table 5). In the following T1, C and T2 periods, the meteorological conditions were less

favourable to the accumulation of pollutants and indeed $N_{ACC}$ concentration was significantly smaller (a decrease of 44%) than during the AC days (Fig. 3b). Nevertheless, $N_{UFP}$ (with a decrease of 13%) and $N_{Q-CRS}$ (with an increase of 10%) concentrations remained close to the levels observed during the anticyclonic period and clearly were less affected by stagnation/ventilation conditions (Fig. 3c). Statistical analysis using the Kruskal–Wallis test confirms that number concentrations differ significantly among all meteorological regimes ($p < 0.05$), supporting the observed variations discussed above. It is possible that $N_{UFP}$ concentration was sustained by a constant local source (traffic). The source of the Q-CRS particles, either local or background, could not be determined accurately based on the inspection of time trends. An increase in particle number concentration $N_{Q-CRS}$ (up to 0.61 cm$^{-3}$) was observed (Fig. 3) during the C period (7 - 10 February). This behaviour is characteristic of coarse-mode particles, which are commonly associated with primary emissions such as mineral dust and sea salt (Seinfeld and Pandis, 2016). An increase in the number concentration of coarse particles may therefore be indicative of long-range or regional transport events, when enhanced advection can bring dust or sea-salt aerosols into the measurement area (Textor et al., 2006), as already observed in the Fairbanks area since the early 90s' (Shaw, 1991a,b).

To investigate source contributions in greater detail, we applied a footprint-based analysis. Specifically, we used bivariate polar plots (pollution roses; Fig. A1) to examine the dependence of UFP, ACC, and Q-CRS particle number concentrations on wind speed and wind direction at the UAF Farm site, stratified by synoptic regime. This approach provides a clear visualization of how different meteorological conditions modulate source impacts. Under anticyclonic conditions, the pollution roses consistently point to downtown Fairbanks as the dominant source region for all three particle size classes. The highest concentrations occurred at low wind speeds, particularly for air masses arriving from the S–SE sector, indicative of stagnant conditions that favour the accumulation of locally emitted particles. An additional component is associated with air masses transported from the W–NW sector under stronger winds. These flow conditions are linked to rural areas outside the Fairbanks basin or to traffic sources in the Goldstream (a tributary valley of the Tanana basin). This source apportionment is consistent with the behaviour of BTEX compounds (benzene, toluene, ethylbenzene and xylene) measured during the ALPACA campaign. In particular, UFP concentrations showed a strong association with toluene (Fig. A2a), the most reliable traffic tracer, especially under anticyclonic conditions when calm winds from the S–SE favoured the accumulation of urban emissions from downtown Fairbanks. By contrast, Q-CRS particles exhibited only a weak relationship with BTEX (not shown here), reflecting the contribution of sources other than local traffic. ACC and Q-CRS particles displayed (Fig A2b,c) a better correspondence with benzene, which in the Fairbanks area is influenced not only by traffic but also by biomass burning and regional background transport. Episodes of enhanced ACC and Q-CRS concentrations not mirrored by BTEX further support the presence of additional, non-traffic sources affecting these particle size classes. Atmospheric stability tends to promote the accumulation of pollutants at the surface level. Nevertheless, during the ALPACA campaign at the UAF site, the periods of most pronounced surface temperature gradients did not always correspond to aerosol concentration peaks (Fig. A3) in contrast to the parallel measurements carried out at a downtown site where a good correlation was found between the extent of surface temperature gradients and PM$_{2.5}$ concentrations (Simpson et al., 2024). This can be partly explained by the limited presence of significant direct emission sources within the source area footprint of the UAF Farm site with respect to the total emissions in

the Fairbanks basin. Aerosol concentrations are influenced by the transport from the neighbouring residential districts (with heating and transportation) and/or from the power plants in the extended area. Therefore, if at a first instance atmospheric stability favours the rise of the concentrations of pollutants, too strong stratifications may induce near-source segregation and inhibit transport from downtown to the suburbs (Brett et al., 2025).

The diurnal variability in the aerosol concentration in all three size ranges (Fig. 4), exhibiting a maximum in daytime hours, must be put in relationship with anthropogenic sources (e.g. traffic, heating). A secondary maximum observed for $N_{ACC}$ particles in the evening hours can be due specifically to residential heating sources, while the $N_{UFP}$ particle concentration was very reduced outside the hours of intense traffic. Ketcherside et al. (2025) indicates that residential heating in Fairbanks has a complex diel pattern with a clear maximum in evening hours. The $N_{Q-CRS}$ exhibited a greater variability pointing to a more varied pattern of sources. The diurnal variability during the AC period is more pronounced possibly because of the more variable wind conditions (Fig. 4d - red arrows). In contrast, in the C period, exhibiting reduced daily wind variability, the aerosol diel trends are also less pronounced (Fig. 4d - blue arrows). Nevertheless, in the cyclonic period there remains a daily pattern in the $N_{UFP}$ with a maximum between 8:00 and 18:00 LT that followed the hydrocarbon-like organic aerosol (HOA) trend (with an Aerosol Mass Spectrometer measurements) obtained by Ijaz et al. (2024) at the downtown site CTC. Thus, even under cyclonic conditions, the traffic emission profile is well reflected in the $N_{UFP}$.

The highest concentrations occurred during persistent anticyclonic periods with strong surface-based inversions, weak winds, and low mixing heights, which favour the accumulation of locally emitted particles. By contrast, during frontal passages and enhanced mixing, number concentrations dropped by an order of magnitude. These results therefore fall within the expected range for Fairbanks wintertime conditions and reflect the strong modulation of aerosol concentrations by meteorology.

### 3.3 Particle Turbulent Fluxes

The turbulent flux statistics of ultrafine ($F_{UFP}$), accumulation ($F_{ACC}$) and quasi-coarse ($F_{Q-CRS}$) particles are listed in Table 1. The campaign-average $F_{UFP}$ was 624 cm$^{-2}$ s$^{-1}$ (median 237 cm$^{-2}$ s$^{-1}$), specifically, the measurement site behaved slightly prevalently as an emission area (positive flux for 58% of the quality-assured cases) for particles in this size range in all the periods of the campaign (Fig. 5a).

Further, the average value of $F_{ACC}$ was 1.74 cm$^{-2}$ s$^{-1}$ (median 0.56 cm$^{-2}$ s$^{-1}$) being substantially bidirectional or slightly positive (Fig. 5b) and greater (as absolute values) in the AC period with respect to the following phases C (3.2 times) and T$_2$ (1.4 times) of the campaign (Table 4 and Table 5). Finally, the mean of $F_{Q-CRS}$ was 0.009 cm$^{-2}$ s$^{-1}$ (median 0.014 cm$^{-2}$ s$^{-1}$). Overall, in mean and median values, quasi-coarse mode fluxes were positive and were, on average as absolute values, very small (Fig. 5c). Similarly, $F_{ACC}$ are positive in 51% of validated cases, while $F_{Q-CRS}$ is positive for 52% of the events.

The diurnal variability in the aerosol emission (positive) fluxes in UFP, ACC and Q-CRS size range (Fig. 6), exhibited a peak value between 6:00 and 9:00 LT in the morning, pointing to a traffic pollution source during the rush hours. A secondary maximum was observed for $F_{UFP}$ and $F_{ACC}$ in the evening hours (around 18:00 LT) that can be due also to the evening rush hour (Fig. 6a and b). In general, the presence of a correspondent peak in the $F_{UFP}$ and $F_{ACC}$ mode particles indicates that a

common emissive source was present in the area in those days (Ketcherside et al., 2025). The road traffic, both very close to the site and/or further away from the highway at north-west, could be the main source in the area. $F_{Q-CRS}$ exhibited large variability with some episodic peak values in the morning between 5:00 and 9:00 LT followed by near null values in the late evening and at night (Fig. 6c). This variability points to a more varied pattern of sources. Among them, a potential particle re-emission from the snow surface through physical processes such as wind-driven resuspension, sublimation, and snow metamorphism (Hagenmuller et al., 2019). Periods with stronger or more variable winds were associated with increased variability in $F_{Q-CRS}$. Selecting $F_{Q-CRS}$ by wind speed, the mean flux was 0.02 cm$^{-2}$ s$^{-1}$ under high wind conditions (> 2.35 m s$^{-1}$), while it was close to zero and slightly negative ($-3.92\times10^{-5}$ cm$^{-2}$ s$^{-1}$) under low wind speeds (< 2.35 m s$^{-1}$), supporting the hypothesis of wind-driven resuspension of particles from the surface contributes to the observed fluctuations in deposition fluxes. On the other hand, during the cyclonic period particle fluxes in all three size modes result lower, especially for the ACC and Q-CRS modes, representing 34% and 28% of the corresponding ones in AC regimes, respectively (Fig. 6). Net particle fluxes remained consistently positive on average, for UFPs during the cyclonic period. To better disentangle the role of sources and boundary-layer processes, we included the diurnal pattern of vertical turbulence intensity ($\sigma_w$) in Fig. 6. The results show that peaks in fluxes often coincide with enhanced turbulence, indicating that surface-layer mixing contributes to the observed diurnal variability alongside traffic emissions and deposition. These observations help separate source-driven signals from processes related to deposition and turbulent transport (Fig. 6).

The comparison of the diurnal cycle between the two synoptic regimes further highlights the role of large-scale circulation in controlling particle concentrations and exchange processes. During anticyclonic conditions, fluxes remain consistently higher throughout the day for the ACC particles, and on average higher between 0:00 and 6:00 LT for UFP and Q-CRS particles, during a time of the day when wind speed and TKE are enhanced during the AC period. The larger fluxes observed in the anticyclonic period with respect to the cyclonic conditions therefore not only reflects the overall increase in particle number concentrations but also suggests more favourable micrometeorological conditions for upward and downward transport, such as enhanced turbulence and stronger coupling between the surface and the boundary layer.

### 3.4 Size-segregated particle deposition velocity $V_d$

As discussed in Sect. 3.3, particle fluxes in all three size classes exhibited bidirectional behaviour, and deposition events were consistently observed during the campaign. To characterize particle transfer processes between the atmosphere and the snow-covered surface, we utilized the exchange velocity ($V_{ex}$) as defined in Equation (1). Over the full measurement period, the median $V_{ex}$ values were -0.19 mm s$^{-1}$ ($p_{10-90}$: -4.38 to 1.52 mm s$^{-1}$) for ultrafine particles (UFP), -0.09 mm s$^{-1}$ ($p_{10-90}$: -1.20 to 0.80 mm s$^{-1}$) for the accumulation mode (ACC), and -0.42 mm s$^{-1}$ ($p_{10-90}$: -6.49 to 5.93 mm s$^{-1}$) for the quasi-coarse size range (Q-CRS). Here, the $p_{10-90}$ indicates the interquartile range between the 10$^{th}$ and 90$^{th}$ percentiles. To specifically investigate dry deposition, only positive values of $V_{ex}$ were considered, enabling the analysis of deposition velocity ($V_d$) (Table 2, Fig. A4). The resulting median $V_d$ values were 0.61 mm s$^{-1}$ ($p_{10-90}$: 0.07 to 3.71 mm s$^{-1}$) for UFP, 0.39 mm s$^{-1}$ ($p_{10-90}$: 0.07 to 1.34 mm s$^{-1}$) for ACC, and 2.56 mm s$^{-1}$ ($p_{10-90}$: 0.72 to 8.73 mm s$^{-1}$) for Q-CRS.

The observed dry deposition velocity in this work is in good agreement with previous measurements of particle dry deposition onto snow and ice surfaces in polar regions. However, direct comparisons remain challenging due to variations in instrumentation and methodologies across studies (e.g., passive sampling, gradient method, eddy covariance). Additionally, few measurements have been conducted in snowy suburban/urban Arctic environments, with most available data originating from remote polar sites or continental interiors. Nonetheless, our EC-based dry deposition velocities measured during ALPACA show good agreement (Pearson $R^2 = 0.9915$, mean difference 0.26 mm s$^{-1}$) with those reported for a snow-covered site in Svalbard (Donateo et al., 2023), which employed a comparable experimental setup (Fig. 7a).

Further, a fair agreement with observations from Contini et al. (2010) (in Antarctica) for total particles >10 nm (median $V_d$ = 0.65 mm s$^{-1}$), and with results from Held et al. (2011) (on the Arctic ice pack) reporting a $V_d$ of 0.59 mm s$^{-1}$ for 0.032 µm particles. Comparisons with Nilsson and Rannik (2001) show values of 1.4 mm s$^{-1}$ at 0.02 µm and 0.25 mm s$^{-1}$ at 0.065 µm, while Ibrahim et al. (1983) (onto snow) observed a $V_d$ of 0.97 mm s$^{-1}$ at 0.70 µm, which aligns reasonably with our measurement of 1.30 mm s$^{-1}$ at 0.61 µm.

Comparing our observations to predictive dry deposition models (Fig. 7b) reveals that the model proposed by Slinn (1982) matches the measured particle size dependency reasonably well, predicting a $V_d$ minimum around 0.1 - 0.2 µm. This suggests that on snow surfaces, mechanisms such as interception for larger particles and Brownian motion for smaller particles dominate aerosol deposition. Conversely, our data indicate that existing parameterizations, including those by Zhang et al. (2001) and Pleim and Ran (2011), may underestimate particle deposition rates between 0.5 and 3 µm, a discrepancy also noted by Donateo et al. (2023). Notably, while Zhang et al. (2001) and Pleim and Ran (2011) predict minimum $V_d$ values at 1.4 µm and 2.4 µm respectively, both Slinn (1982) and our observations point to a minimum around 0.15 µm.

Finally, the size dependence of $V_d$ across the spectrum was fitted using a fourth-order polynomial function of the particle diameter ($d_p$), capturing the characteristic decline in deposition efficiency within the Aitken mode (50–150 nm). Fitting coefficients, the $R^2$ value, and the RMSE of the fit are reported in Table 3.

**3.5 Particle deposition velocity and the friction velocity u\***

The deposition velocity ($V_d$) is strongly influenced by turbulence intensity, typically characterized through the friction velocity (u*), with higher turbulence enhancing particle fluxes (Sievering, 1967; Grönholm et al., 2007; Vong et al., 2010; Ahlm et al., 2010). During our measurement period, the average friction velocity (Fig. A6a) was 0.10 m s$^{-1}$, a value comparable to that reported by Wendler (1969) for snow surfaces (0.07 m s$^{-1}$). In agreement with previous findings (Nilsson and Rannik, 2001; Contini et al., 2010; Pryor et al., 2013; Donateo et al., 2023), our observations indicate that $V_d$ systematically increases with u* across all particle size ranges. Specifically, when u* exceeds 0.10 m s$^{-1}$, the relationship between deposition velocity and friction velocity is approximately linear. Our data confirm this linear trend (Fig. 8a), with slopes varying as a function of particle size: m = 0.008 ($R^2 = 0.96$) for UFP, m = 0.002 ($R^2 = 0.91$) for ACC, and m = 0.023 ($R^2 = 1$) for Q-CRS.

To enhance comparability across studies, we normalized the deposition velocity by friction velocity ($V_n = V_d/u*$), following a common approach (Table 3). The median $V_n$ values across the full period (Fig. A5) were 9.9·10$^{-3}$, 4.7·10$^{-3}$, and 36.5·10$^{-3}$ for

UFP, ACC, and Q-CRS, respectively. A good agreement was found with the measurements by Duann et al. (1988), for the lower portion in the accumulation mode, with a mean value of $V_n = 0.006$. As for $V_d$, an analogous fit for $V_n$, according to the equation (3), has been performed (Fig. 8b). The data driven fit resulted in a similar goodness and respective fitting coefficients are reported in Table 3. Studies on size segregated particles deposition velocity conducted at Ny Ålesund by Donateo et al. (2023) also show a strong linear relationship between $V_d$ and u*, and similarly to Fairbanks, the fitting slopes (and to a lesser extent the correlation coefficients) vary considerably with particle size, with some differences between the two measurement sites. However, the comparison of the observed normalized deposition velocity ($V_n$) between Fairbanks and Ny-Ålesund sites reveals some differences. Specifically, the medians are statistically different, as determined by a Wilcoxon-Mann-Whitney test (p-value $< 0.05$), within the size range of 0.54 to 0.89 μm (large accumulation mode). This different deposition behaviour in the accumulation mode could be due to different properties in snow cover (roughness and porosity), different local atmospheric conditions (atmospheric stability, turbulence intensity, or relative humidity), or a different particle chemical composition (some particles might be more hygroscopic or have different densities) (Seinfeld and Pandis, 2016). Finally, in dry and cold polar conditions, particles can acquire electrostatic charges, affecting their deposition (Tkachenko and Jacobi, 2024). If humidity and solar radiation conditions differ between the sites, the surface charge of the particles might change, modifying the deposition efficiency for intermediate-sized particles.

### 3.6 Relationship between particle deposition velocity and meteorological conditions

Intense negative (depositional) particle fluxes were sometimes observed in January and on the first days of February, during the anticyclonic period (AC) (Fig. 5). This can be put in relation with the higher particle loadings observed during the stagnating conditions observed in this first phase of the campaign (Fig. 3). However, after normalization for particle concentration, some differences between the periods of the campaign can still be observed, showing smaller $V_d$ values in all size classes for the C period with respect to the others (Table 4 and Table 5), even if a statistical analysis by Kruskal–Wallis tests indicates that $V_d$s did not differ significantly among meteorological regimes (p $> 0.05$). Based on the results of Maillard et al. (2022) and Pohorsky et al. (2025), two further meteorological regimes can be considered within the anticyclonic period: conditions with strong negative radiative imbalance ($< -25$ W m$^{-2}$) and surface winds (wind speed at 2 m $> 1$ m s$^{-1}$), labelled hereafter as $AC_a$, and a second regime characterized by a weak negative radiative budget ($> -25$ W m$^{-2}$) or calm conditions (wind speed at 2 m $< 1$ m s$^{-1}$), referred to hereafter as $AC_b$. Other conditions observed during the AC period, namely the strong advection event of 28 Jan 00:00 - 15:00 LT were excluded from the statistical analysis reported for $AC_a$ and $AC_b$ in Table 4. The mean friction velocity (u*) was larger during the anticyclonic period (on average 0.13 m s$^{-1}$), while the lower value was measured during the cyclonic period C (on average 0.09 m s$^{-1}$). It is worth noting median u* (0.13 m s$^{-1}$) and TKE (0.13 m$^2$ s$^{-2}$) was higher in T2 with respect to AC, but the downward sensible heat flux was greater during AC both as mean and median value. It can be put in relation with the steeper temperature surface gradient and higher wind speed at the surface in the period. Statistical analysis using Kruskal–Wallis tests followed by post-hoc pairwise comparisons indicates that deposition u*, TKE and H differ significantly among several meteorological regimes (p $< 0.05$). At the same time, such gradient was the highest

(4.72 °C) in the T1 period, but concomitantly to a limited turbulence activity ($u^* < 0.1$ m s-1), so that the downward sensible flux was smaller than during AC. Such characteristics of surface fluxes during the anticyclonic period are more prominent under conditions of combined strong surface radiative cooling and moderate to high surface winds ($AC_a$) link to SCF presence, when turbulence near the surface and steep temperature gradients resulted into strong downward sensible heat fluxes (-7.12 W m$^{-2}$ on average). Table 4 shows that the $V_d$ values for all three particle classes were always higher when surface cooling was strong, and winds were sustained ($AC_a$) with respect to $AC_b$ conditions. Interestingly, particle deposition is enhanced in conditions outside the main pollution events, which occur when atmospheric circulation is from downtown Fairbanks at low wind speeds (1.30 m s$^{-1}$) especially at daytime when the net surface radiation is around zero ($AC_b$). Conversely, background conditions characterised by sustained winds at only moderate aerosol concentration levels turn out to be more favourable to particle deposition. The fact that accumulation mode concentrations ($N_{ACC}$) were larger during $AC_b$ conditions (178.77 cm$^{-3}$) than during the more stable $AC_a$ (134.84 cm$^{-3}$) ones, but depositional fluxes ($F_{ACC}$) exhibit an opposite behaviour, shows the importance of meteorological factors in determining the magnitude of aerosol depositions in this environment.

To further investigate the relationship between surface layer meteorology and particle depositional fluxes, Fig. 9 shows the trends for selected weather and EC parameters during typical days of anticyclonic conditions (29$^{th}$ January – 1$^{st}$ February). Solar irradiation and intermittent clouds were responsible for short periods of net positive surface radiation balance, but a net deficit of at least - 25 W m$^{-2}$ was observed for most of the time ($AC_a$, no colour bars). Winds were mostly north-westerly with speed exhibiting an irregular trend (Fig. 9b). Nevertheless, under the daytime $AC_b$ conditions (yellow bars), wind speeds sometimes decreased, with wind direction turning to SE in two days out of four (Fig. 9b). For these two days, the southeasterly air masses showed some wind shear and were characterized by very low wind speeds in the middle of the day. Turbulence indicators (represented by $u^*$ and TKE, here) exhibited a more consistent temporal trend and did not closely follow wind speed. In particular, after sunset turbulence increased at a slower rate with respect to wind speed and it reached a maximum between midnight and 11:00 LT in the morning. In the same hours, a maximum of sensible heat flux was observed (Fig. 9d). Therefore, during the night between 29$^{th}$ and 30$^{th}$ January, surface cooling accompanied a growing surface temperature gradient until midnight, followed by several hours during which positive H flux curbed the cooling and reduced slowly the temperature gradient until 7:00 LT when the presence of clouds and, later, the sunrise led to positive radiative budgets. The deposition velocities for UFP, ACC and Q-CRS showed a net increase after midnight concomitantly with $u^*$ and H, and similarly strongly decreased after 11:00 LT in the morning (Fig. 9d). It is worth noticing that in Fairbanks in the polar winter, the daytime solar irradiation reduces surface cooling, but it is not enough to set up turbulence by increasing buoyancy: on the contrary, in the daytime a transition to an intermittent regime of the turbulence was observed ($\zeta$ was about 1.6). EC observations stopped after 30 Jan midnight, while during the night between 31$^{st}$ January and 1$^{st}$ February similar processes were observed with respect to 29$^{th}$ - 30$^{th}$ January, with an increase of $u^*$ and H after midnight, although with a more complex evolution of the surface temperature gradient.

The deposition velocity in the ACC mode reached relative minimum values in periods of relatively higher temperature and large $\Delta T$. This could indicate an effect of $\Delta T$ on $V_d$ net of the $u^*$ variability, and can provide explanation why during the T1

period, which was characterized by very strong ΔT, $V_d$ was also high when compared to the following C and T2 period, in spite of the fact that turbulence conditions were not favourable to enhance deposition velocities (Table 2). We can speculate that near-surface gradients in temperature enhance particle deposition through phoretic forces (Farmer et al., 2021). Further investigations are needed to assess the importance of phoretic processes with respect to strong stability preventing particle deposition over the snowfields of Fairbanks.

**3.7 Effect of the boundary layer vertical structure on the particle fluxes**

During the polar winter in continental areas like the Fairbanks basin, atmospheric stability is normally associated with thermal inversions and the presence of aerosol layers. During the ALPACA campaign, fine-resolved atmospheric profiling was carried out using tethered balloon flights (helikite) technology with a total of 24 flights between 26[th] January and 25[th] February (see Pohorsky et al., 2025, for a detailed overview). Typically, a surface-based thermal inversion was observed in the first 100 m a.g.l., while elevated inversions were occasionally observed between 200 and 400 m a.g.l. The concentrations of aerosol and trace gases were generally higher below the SBI, while the elevated inversions could mark the transition between moderately polluted to very clean conditions above. Nevertheless, elevated layers of pollution were frequently observed above the SBI, up to 300 m a.g.l. associated with power plants plumes (Brett et al., 2025). Close to the surface, wind friction could develop into a surface mixing sub-layer connected to the SBI by an intermediate layer of very steep temperature gradient. Such a dynamical feature of the mixing layer depends on wind speed, and it was observed on about half of the flights performed during the ALPACA campaign (Pohorsky et al., 2025). The results from two helikite flights (n. 4 and n. 6 in Pohorsky et al., 2025) carried out in the same time span of the surface measurements are discussed here (Figs. 10 and 11). These two flights were performed during the anticyclonic period, when atmospheric stability favoured the accumulation of pollutants at the surface but at the same time also stratification, hence reducing vertical fluxes. During both flights significant particle $V_d$ values were observed, in spite of the evident features of multiple stratification determined by the helikite profiles (Fig. 10a and b). The first flight (n. 4) was carried out in the early morning hours, started measuring well before dawn. Initially, a steep surface temperature gradient and a SBI at around 40 m a.g.l. could be observed, along with multiple aerosol layers with a maximum in concentrations just below the SBI (Fig. 10c and d). Although the surface layer could be probed only during fast ascending or descending, two out the three profiles obtained before 8:30 LT showed features of a 5-to-10 m thick mixing sublayer, indicating that wind friction generated turbulence and mixing very close to the ground. As the EC station was placed at 11 m a.g.l., these measurements could only marginally capture such turbulent flux (Fig. 10a and c), which explains the quite irregular trend in the measured deposition velocity. Yet, the helikite profiling demonstrates that heat and particle fluxes could be generated mechanically very close to the surface even in the presence of very steep surface temperature gradients. The last two profiles were collected between 9:30 and 10:30 LT at a time of the day when surface radiation budget was increasing and surface winds and u* decreasing, eventually transitioning into an $AC_b$ regime. The particle concentration increased dramatically in the surface layer while the temperature gradients were greatly reduced. Although the flat temperature profile might point to surface mixing, the particle vertical distribution exhibited multiple fine layering witnessing extensive

stratification (Brett et al., 2025). Indeed, at this time of the day, turbulence was much reduced and despite the increased aerosol concentration (probably associated with surface level transport from downtown), such enhanced aerosol loads could not find their way to the ground by deposition. Interestingly, the wind field during the SE flows from downtown show decoupled wind directions between 2 and 11 m (Fig. 9b), which can indicate the occurrence of extensive stratifications hindering vertical fluxes (Lan et al., 2022).

The second flight (n. 6) was instead carried out late in the evening and extended over several hours at night, with surface temperature progressively becoming colder. An SBI was found at 35 m a.g.l., although several elevated inversions were found up to 250 m a.g.l. (Fig. 11a and b). This made the aerosol concentration - higher in the surface layer - to decrease stepwise down to a very clean layer above 100 m a.g.l. (Fig. 11c). The aerosol vertical structure in the surface layer could only be probed three times and for a few minutes but showed a rather constant vertical profile extending below the SBI down to the surface. This, together with the rather flat temperature profiles below an elevation of 20 - 25 m a.g.l. (Fig. 11a and b), point to an efficient mixing in a surface sublayer (Pohorsky et al., 2025). These results provide further evidence that mechanically generated turbulence can be responsible for heat and particle vertical fluxes in the lowest ~20 m of the atmosphere even in the condition of strong atmospheric stability (Maillard et al., 2024).

**4 Conclusions**

The study examined the relationship between particle fluxes, meteorological conditions, and air quality at a sub-urban site in Fairbanks during the 2022 ALPACA campaign. The measurement site acted predominantly as a source, showing mainly positive (emission) fluxes for the three particle modes observed (ultrafine - UFP, accumulation - ACC, and quasi-coarse Q-CRS) and more consistently for UFP. A clear fingerprint of local traffic sources was identified in the diurnal cycle of the emission fluxes for at least the UFP size range. However, the inspection of the diel variability of particle air concentrations suggests that advection from downtown was an important source of aerosols for this site along with nearby residential areas, the airport and UAF activities. The median deposition velocity values were 0.61, 0.04 and 8.73 mm s$^{-1}$, for UFP, ACC and Q-CRS, respectively, with Q-CRS particles exhibiting the largest deposition rates. Particle fluxes showed a marked dependence on meteorological conditions with wind-driven turbulence directly impacting aerosol deposition, particularly in periods of increased surface cooling and strong temperature inversions (and very reduced buoyancy). The relationship between friction velocity and particle deposition velocity aligned with that observed in previous studies, reinforcing the notion that dry deposition is significantly influenced by atmospheric turbulence (i.e. shear stress). During the anticyclonic conditions that dominated during the first part of the campaign, stagnant air masses contributed to the accumulation of pollutants near the surface, but the highest levels of particle concentrations did not correspond to the maximum depositional fluxes. On the contrary, the advection of polluted air from downtown was generally associated to a stratified atmosphere and reduced turbulence at the surface, while air flows from north-west bringing only moderate concentrations of pollutants were responsible for peaks in particle deposition in condition of enhanced turbulence. During periods of strong radiative cooling, enhanced

thermal gradients between the air and the snow surface likely contributed to increased particle downward fluxes. This may suggest that traditional parameterizations of dry deposition in atmospheric models could underestimate the role of thermophoresis in Arctic environments that often take place across sharp thermal gradients over a cold surface. While our results provide robust evidence of the influence of atmospheric stability on aerosol fluxes within the context of this short-term campaign, the impracticability to perform flux measurements right at the top of the mixing layer in all weather conditions limits our ability to fully constrain the vertical particle budget. This limitation should be considered when interpreting the relative contributions of surface exchange versus entrainment, divergence or dilution processes, particularly under well-mixed conditions. Nevertheless, the quantitative determination of size-segregated particle depositions presented in this study offers valuable insights to help fill the measurement gap for aerosol deposition processes in the Arctic as well as to reducing the uncertainties in the parameterizations used to represent such processes in regional and global atmospheric transport models and Earth system models. In respect to the vulnerability of Arctic urban environments, a better quantitative understanding of dry deposition is expected to support more accurate predictions of particulate matter sinks and atmospheric lifetime, as well as of the fluxes of contaminants (e.g. black carbon) to the snowpack and to Arctic ecosystems. A follow-up study will leverage the results on particle deposition velocities determined in the present study to assess the effects of atmospheric dry deposition on snowpack chemical composition in Fairbanks, providing another contribution of the ALPACA research on the air pollution impacts in cold climates.

**Appendix A**

*Data Availability*: Final data from the study will be available to the scientific community through the ALPACA data portal hosted by Arcticdata.io (https://arcticdata.io/catalog/portals/ALPACA).

*Author contribution*: A.D., S.D. planned the experimental design and guided the research. A.D., G.P., F.S., M.B. collaborated on data collection and post-processing. A.D., G.P., and S.D. carried out the analysis presented in this paper and drafted the manuscript with contributions from all co-authors. A.D., G.P. developed the code used to analyse the data. A.D. and S.D. managed and provided the Italian funding projects. R.P. and A.B. performed the helikite measurements with the assistance of F.S., G.P., M.B., S.B., B.B. and G.J.F. R.P. performed data curation and carried out the data analysis for the helikite with support from J.S. W.R.S., K.S.L., J.S., G.J.F. and S.R.A. initiated and designed the ALPACA study and obtained funding. All authors reviewed and edited the manuscript, cooperating to interpret the results, wrote, read, commented, and approved the final manuscript.

*Competing interests*: The authors declare no competing interests.

*Financial Support*: This work has been funded and conducted in the framework of the Joint Research Center ENI-CNR - "Aldo Pontremoli" within the ENI-CNR Joint Research Agreement, WP1 "Impatto delle emissioni in atmosfera sulla criosfera e sul

cambiamento climatico nell'Artico". Further, this work was financed by the PRA (Programma di Ricerche in Artico) 2019 program (project A-PAW). This work received funding from the Swiss Polar Institute (Technogrant 2019) and the Swiss National Science Foundation (grant no. 200021_212101). J.S holds the Ingvar Kamprad Chair for Extreme Environments Research sponsored by Ferring Pharmaceuticals. N.B., B.B., E.D., S.B. and K.S.L. acknowledge support from the Agence National de Recherche (ANR) CASPA (Climate-relevant Aerosol Sources and Processes in the Arctic) project (grant no. ANR-21-CE01-0017), and the Institut polaire français Paul-Émile Victor (IPEV) (grant no. 1215) and CNRS-INSU programme LEFE (Les Enveloppes Fluides et l'Environnement) ALPACA-France projects.

*Acknowledgements:* We would like to thank the ALPACA community for the organization and research performed in Fairbanks. The ALPACA project is organized as a part of the International Global Atmospheric Chemistry (IGAC) project under the Air Pollution in the Arctic: Climate, Environment and Societies (PACES) initiative with support from the International Arctic Science Committee (IASC), the National Science Foundation (NSF), and the National Oceanic and Atmospheric Administration (NOAA). We express our gratitude to the University of Alaska Fairbanks and the Geophysical Institute for their support. More specifically, we thank Gilberto J. Fochesatto, William R. Simpson and Jinqiu Mao for their help on site.

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

**Table 1: Statistical description for particle number concentration (N) and turbulent particle number flux (F), segregated for UFP, ACC and Q-CRS particle size ranges. $d_p$ represents the arithmetic mean diameter of each size range. $p_{10}$ and $p_{90}$ represent the 10th and 90th percentiles.**

| | N ($cm^{-3}$) | | | F ($cm^{-2} s^{-1}$) | | |
|---|---|---|---|---|---|---|
| | UFP | ACC | Q-CRS | UFP | ACC | Q-CRS |
| $d_p$ (μm) | 0.013 | 0.47 | 1.80 | 0.013 | 0.47 | 1.80 |
| $p_{10}$ | 2,650 | 35.93 | 0.17 | -2,032 | -6.28 | -0.19 |
| mean | 16,849 | 91.76 | 0.35 | 624 | 1.74 | 0.009 |
| std.dev | 16,348 | 65.20 | 0.21 | 3,536 | 11.96 | 0.18 |
| std.err | 519 | 2.06 | 0.007 | 126 | 0.41 | 0.007 |
| median | 12,767 | 75.57 | 0.29 | 237 | 0.56 | 0.014 |
| $p_{90}$ | 33,263 | 167 | 0.59 | 3,867 | 9.75 | 0.19 |

**Table 2: Statistical parameters for the deposition velocity $V_d$ and deposition velocity normalised for the friction velocity $V_n$ calculated on the whole measurement period, separating the UFP, ACC and Q-CRS particle size modes. $d_p$ represents the arithmetic mean diameter of each size range. $p_{10}$ and $p_{90}$ represent the 10th and 90th percentiles.**

| | $V_d$ (mm s$^{-1}$) | | | $V_n$ (mm s$^{-1}$) | | |
|---|---|---|---|---|---|---|
| | **UFP** | **ACC** | **Q-CRS** | **UFP** | **ACC** | **Q-CRS** |
| **$d_p$ (µm)** | 0.013 | 0.47 | 1.80 | 0.013 | 0.47 | 1.80 |
| **$p_{10}$** | 0.07 | 0.07 | 0.72 | 1.30E-03 | 1.00E-03 | 1.07E-02 |
| **mean** | 1.47 | 0.59 | 3.74 | 1.76E-02 | 7.10E-03 | 4.57E-02 |
| **std.dev** | 2.68 | 0.72 | 3.48 | 2.77E-02 | 9.50E-03 | 3.77E-02 |
| **std.err** | 0.15 | 0.04 | 0.19 | 1.50E-03 | 5.00E-04 | 2.0E-03 |
| **median** | 0.61 | 0.39 | 2.56 | 9.90E-03 | 4.70E-03 | 3.65E-02 |
| **$p_{90}$** | 3.71 | 1.34 | 8.73 | 4.17E-02 | 1.92E-03 | 8.87E-02 |

**Table 3: Polynomial fitting coefficients [s$^{-1}$] and goodness of fit for each observed curve.**

|       | $p_1$ | $p_2$ | $p_3$ | $p_4$ | $p_5$ | $R^2$ | RMSE  |
|-------|-------|-------|-------|-------|-------|-------|-------|
| $V_d$ | 0.29  | -1.40 | 2.28  | 1.37  | -0.26 | 0.91  | 1.89  |
| $V_n$ | 0.01  | -0.05 | 0.09  | -0.01 | 0.001 | 0.92  | 0.023 |

**Table 4: Mean, median and interquartile range of selected meteorological parameters, N particle concentrations, and F particle flux for the anticyclonic period (AC) of the campaign, as well as for the AC$_a$ and AC$_b$ regimes (see text). Q1 and Q3 represent the 1$^{st}$ and the 3$^{rd}$ quartiles.**

| | AC | | | | AC$_a$ | | | | AC$_b$ | | | |
|---|---|---|---|---|---|---|---|---|---|---|---|---|
| | mean | median | Q1 | Q3 | mean | median | Q1 | Q3 | mean | median | Q1 | Q3 |
| ΔT (°C) | 2.32 | 2.15 | 0.57 | 3.72 | 3.16 | 3.55 | 1.83 | 4.45 | 1.46 | 1.46 | 0.39 | 2.20 |
| WS (m s$^{-1}$) @ 2 m | 2.08 | 2.04 | 1.41 | 2.78 | 2.34 | 2.14 | 1.76 | 2.80 | 1.30 | 1.00 | 0.30 | 2.04 |
| WS (m s$^{-1}$) @ 11 m | 2.42 | 2.41 | 1.41 | 3.28 | 2.75 | 2.70 | 1.93 | 3.52 | 1.54 | 1.24 | 0.41 | 2.40 |
| Net rad (W m$^{-2}$) | -21.48 | -30.08 | -36.20 | -6.23 | -35.96 | -34.96 | -40.48 | -31.10 | 4.27 | 2.14 | -6.23 | 18.46 |
| H (W m$^{-2}$) | -6.12 | -3.09 | -9.70 | -0.03 | -7.15 | -4.60 | -9.41 | -0.74 | -1.70 | -0.02 | -1.83 | 1.35 |
| H > 0 | 2.31 | 1.48 | 0.47 | 3.01 | 3.27 | 1.86 | 0.95 | 4.75 | 1.64 | 1.43 | 0.29 | 1.91 |
| TKE (m$^2$ s$^{-2}$) | 0.18 | 0.09 | 0.03 | 0.23 | 0.15 | 0.10 | 0.04 | 0.21 | 0.01 | 0.04 | 0.02 | 0.11 |
| u* (m s$^{-1}$) | 0.13 | 0.09 | 0.04 | 0.19 | 0.13 | 0.09 | 0.05 | 0.17 | 0.09 | 0.05 | 0.03 | 0.10 |
| N$_{UFP}$ (cm$^{-3}$) | 21,337 | 14,208 | 9,415 | 24,595 | 16,325 | 13,392 | 9,577 | 17,703 | 37,170 | 27,738 | 14,225 | 40,929 |
| N$_{ACC}$ (cm$^{-3}$) | 138.64 | 117.56 | 78.16 | 173.15 | 134.84 | 122.63 | 91.63 | 169.29 | 178.77 | 129.69 | 97.23 | 231.14 |
| N$_{Q-CRS}$ (cm$^{-3}$) | 0.41 | 0.33 | 0.26 | 0.42 | 0.35 | 0.31 | 0.25 | 0.40 | 0.54 | 0.38 | 0.29 | 0.75 |
| F$_{UFP}$ (cm$^{-2}$ s$^{-1}$) | 578 | 237 | -720 | 1,803 | 983 | 278 | -587 | 2,204 | -255 | -209 | -1,464 | 1,224 |
| F$_{ACC}$ (cm$^{-2}$ s$^{-1}$) | 2.18 | 0.51 | -4.28 | 5.92 | 2.96 | 1.03 | -3.80 | 7.64 | 0.66 | -0.88 | -4.77 | 3.98 |
| F$_{Q-CRS}$ (cm$^{-2}$ s$^{-1}$) | 0.02 | 0.02 | -0.10 | 0.12 | 0.03 | 0.04 | -0.08 | 0.13 | 0.01 | 0.01 | -0.08 | 0.09 |
| F$_{UFP}$ < 0 | -2,055 | -858 | -2,312 | -324 | -1,846 | -705 | -1,698 | -213 | -2,573 | -1,377 | -3,403 | -720 |
| F$_{ACC}$ < 0 | -7.36 | -4.46 | -7.94 | -1.93 | -7.99 | -4.66 | -8.02 | -1.99 | -6.61 | -4.28 | -6.58 | -1.70 |
| F$_{Q-CRS}$ < 0 | -0.15 | -0.10 | -0.20 | -0.06 | -0.13 | -0.11 | -0.18 | -0.06 | -0.14 | -0.08 | -0.13 | -0.06 |
| V$_d$ UFP (mm s$^{-1}$) | 1.49 | 0.55 | 0.20 | 1.38 | 1.53 | 0.68 | 0.13 | 1.77 | 0.86 | 0.34 | 0.18 | 0.81 |
| V$_d$ ACC (mm s$^{-1}$) | 0.63 | 0.38 | 0.14 | 0.80 | 0.62 | 0.32 | 0.17 | 0.80 | 0.44 | 0.23 | 0.08 | 0.53 |
| V$_d$ Q-CRS (mm s$^{-1}$) | 4.33 | 3.13 | 1.65 | 5.38 | 4.27 | 3.31 | 1.83 | 5.53 | 3.00 | 1.92 | 0.81 | 3.30 |

**Table 5: Same as in Table 4 but for periods T1, C and T2. Q1 and Q3 represent the 1st and the 3rd quartiles.**

| | T1 | | | | C | | | | T2 | | | |
|---|---|---|---|---|---|---|---|---|---|---|---|---|
| | mean | median | Q1 | Q3 | mean | median | Q1 | Q3 | mean | median | Q1 | Q3 |
| $\Delta T$ (°C) | 4.72 | 4.02 | 2.02 | 7.83 | 0.79 | 0.41 | 0.22 | 0.93 | 0.59 | 0.34 | 0.21 | 0.69 |
| WS (m s$^{-1}$) @ 2 m | 2.45 | 2.50 | 1.35 | 3.65 | 1.52 | 1.17 | 0.71 | 2.08 | 2.62 | 2.70 | 1.73 | 3.46 |
| WS (m s$^{-1}$) @ 11 m | 2.54 | 2.05 | 0.91 | 4.43 | 1.93 | 1.50 | 0.92 | 2.65 | 3.08 | 3.21 | 2.10 | 4.16 |
| Net rad (W m$^{-2}$) | -21.51 | -31.58 | -40.12 | -4.62 | -7.93 | 0.27 | -15.34 | 4.83 | -15.38 | -21.17 | -36.61 | 12.85 |
| H (W m$^{-2}$) | -6.67 | -4.77 | -10.89 | -0.42 | -1.38 | -0.61 | -2.92 | 0.68 | -2.47 | -1.86 | -4.59 | -0.49 |
| H > 0 | 8.55 | 3.43 | 0.83 | 7.41 | 1.92 | 2.05 | 0.60 | 3.03 | 2.23 | 1.03 | 0.48 | 3.80 |
| TKE (m$^2$ s$^{-2}$) | 0.10 | 0.05 | 0.02 | 0.16 | 0.08 | 0.05 | 0.02 | 0.09 | 0.14 | 0.13 | 0.06 | 0.21 |
| u* (m s$^{-1}$) | 0.09 | 0.06 | 0.03 | 0.14 | 0.08 | 0.06 | 0.04 | 0.09 | 0.13 | 0.13 | 0.08 | 0.19 |
| $N_{UFP}$ (cm$^{-3}$) | 18,135 | 13,024 | 7,318 | 22,369 | 18,739 | 15,955 | 9,856 | 26,948 | 15,196 | 11,757 | 9,455 | 19,253 |
| $N_{ACC}$ (cm$^{-3}$) | 111 | 88.70 | 77.08 | 111 | 78.71 | 64.86 | 55.31 | 98.48 | 71.53 | 69.94 | 37.99 | 101 |
| $N_{Q-CRS}$ (cm$^{-3}$) | 0.34 | 0.29 | 0.21 | 0.43 | 0.46 | 0.45 | 0.33 | 0.57 | 0.24 | 0.21 | 0.18 | 0.29 |
| $F_{UFP}$ (cm$^{-2}$ s$^{-1}$) | 505 | -86.03 | -1,022 | 2,178 | 323 | 134 | -1,169 | 854 | 943 | 523 | -432 | 1,630 |
| $F_{ACC}$ (cm$^{-2}$ s$^{-1}$) | 8.11 | 2.01 | -1.02 | 10.47 | 0.68 | 0.82 | -1.25 | 2.20 | 1.58 | 1.06 | -2.19 | 3.96 |
| $F_{Q-CRS}$ (cm$^{-2}$ s$^{-1}$) | 0.02 | 0.03 | -0.05 | 0.10 | -0.01 | 0.01 | -0.11 | 0.09 | 0.03 | 0.04 | -0.07 | 0.11 |
| $F_{UFP}$ < 0 | -1,871 | -967 | -2,022 | -379 | -1,421 | -1,525 | -2,118 | -377 | -1,875 | -920 | -2,269 | -432 |
| $F_{ACC}$ < 0 | -5.92 | -2.68 | -9.14 | -1.34 | -3.15 | -1.52 | -3.89 | -0.93 | -3.87 | -2.58 | -5.12 | -1.43 |
| $F_{Q-CRS}$ < 0 | -0.10 | -0.07 | -0.10 | -0.04 | -0.14 | -0.11 | -0.21 | -0.05 | -0.12 | -0.10 | -0.15 | -0.06 |
| $V_d$ UFP (mm s$^{-1}$) | 1.70 | 0.77 | 0.28 | 1.79 | 0.86 | 0.57 | 0.29 | 1.01 | 1.34 | 0.63 | 0.31 | 1.58 |
| $V_d$ ACC (mm s$^{-1}$) | 0.65 | 0.32 | 0.13 | 0.98 | 0.45 | 0.24 | 0.13 | 0.69 | 0.71 | 0.49 | 0.33 | 0.74 |
| $V_d$ Q-CRS (mm s$^{-1}$) | 3.62 | 1.88 | 1.15 | 3.86 | 3.60 | 2.46 | 1.14 | 5.53 | 5.42 | 5.72 | 2.90 | 7.45 |

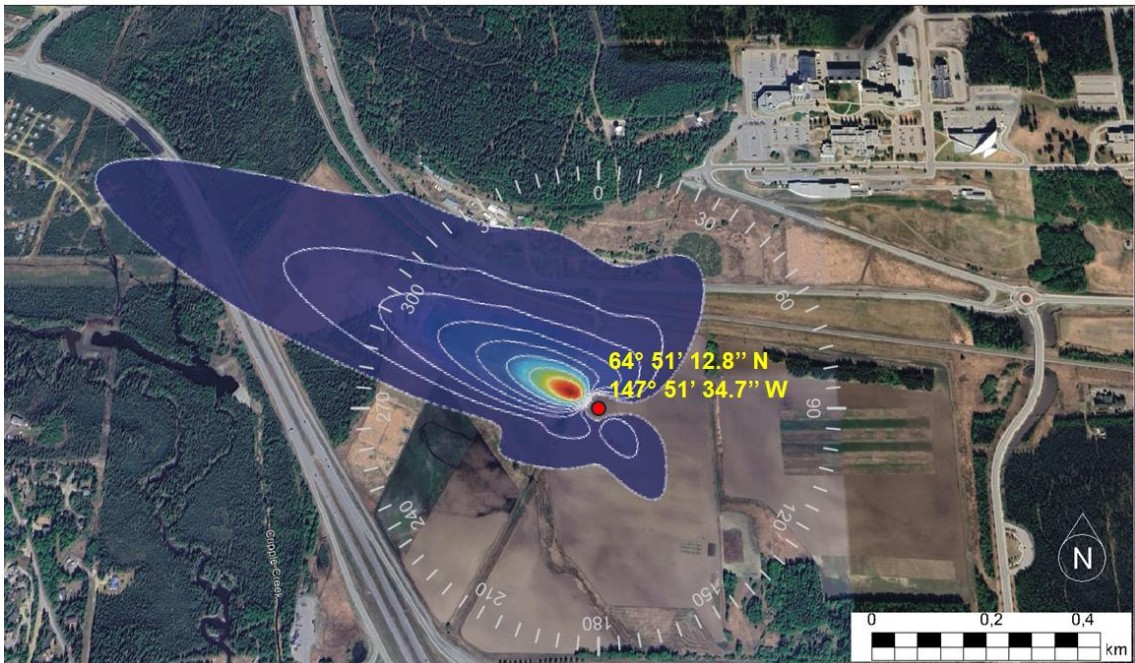

**Figure 1: Location map of the study site: Fairbanks, Alaska, (USA). The red circle indicates exactly the measurement site (the shelter position). The map is obtained from Google Earth ©. In the figure it was also reported the flux footprint area for the measurement setup (Sect. 2.2). Coloured contours in the footprint image represent the relative contribution of each area to the measured flux at the tower, with warmer colours indicating higher contribution.**

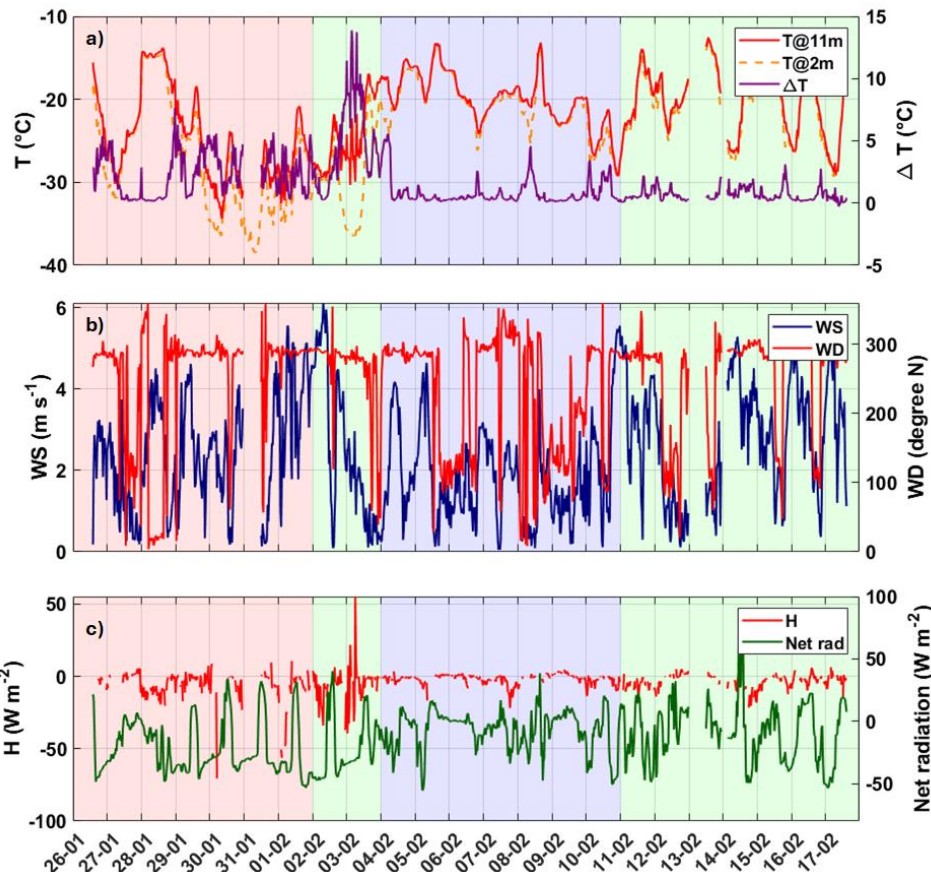

**Figure 2: Time series of the principal meteorological variables measured during the campaign in Fairbanks. (a) air temperature T (°C) at two different heights (2 m and 11 m) and, on the right axis, the temperature difference ΔT, (b) (WS) wind speed (m s⁻¹), and (WD) wind direction (degree N). (c) sensible heat flux H (W m⁻²) and the net radiation (W m⁻²). The colour bands indicate the anticyclonic period AC (red), the cyclonic period C (blue) and the two transition periods T (green).**

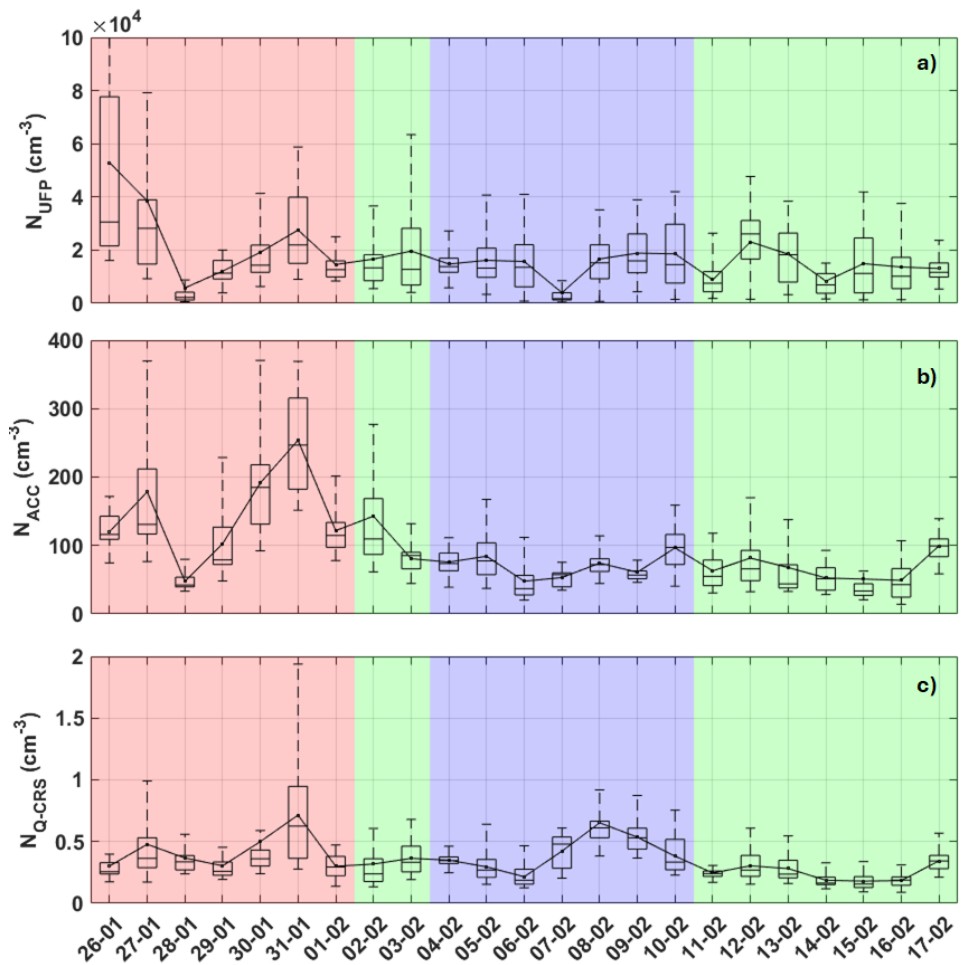

**Figure 3: Daily box plots for (a) $N_{UFP}$, (b) $N_{ACC}$ and (c) $N_{Q\text{-}CRS}$. Boxes represent the 25th and 75th percentiles. Whiskers correspond to $\pm 2.7\sigma$ and 99.3% data coverage. The colour bands indicate the anticyclonic period AC (red), the cyclonic period C (blue) and the two transition periods T (green). The line inside each box indicates the median, while the dots represent the mean values.**

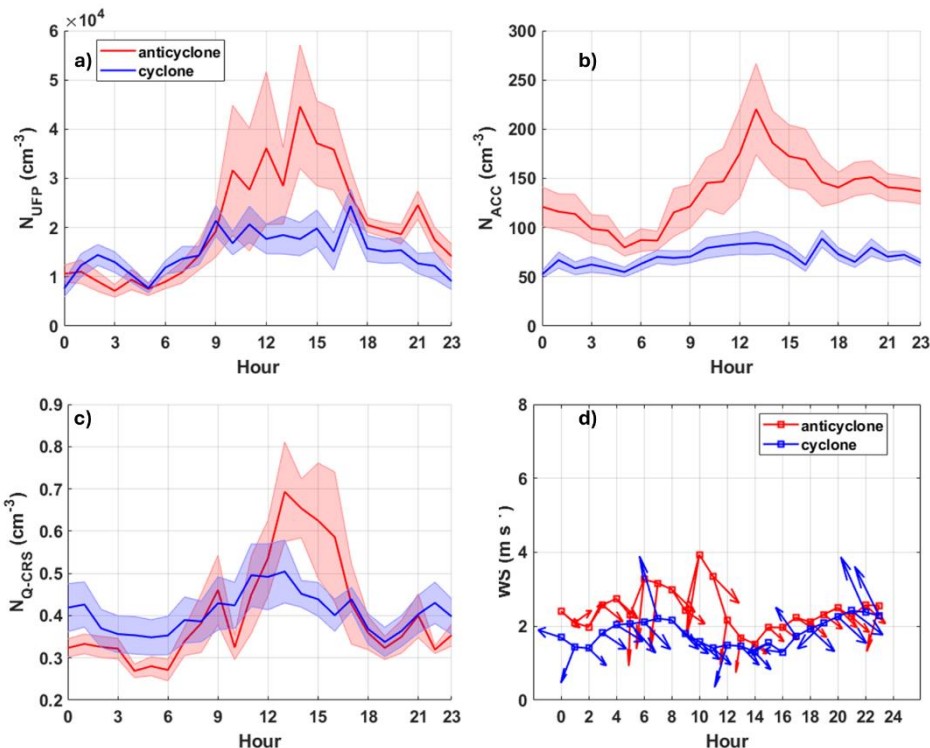

**Figure 4: Diurnal trends (mean) of aerosol concentrations at UAF-Farm for a) $N_{UFP}$, b) $N_{ACC}$ and c) $N_{Q\text{-}CRS}$ particles mode. d) Diurnal trend (mean) for the wind velocity and direction (indicated by the arrows). The diurnal trends are shown both for the anticyclonic (red line) and cyclonic (blue line) period (defined as in Fig. 2). Continuous lines are the mean value. The shadow area represents the standard error. Hour is in standard local time.**

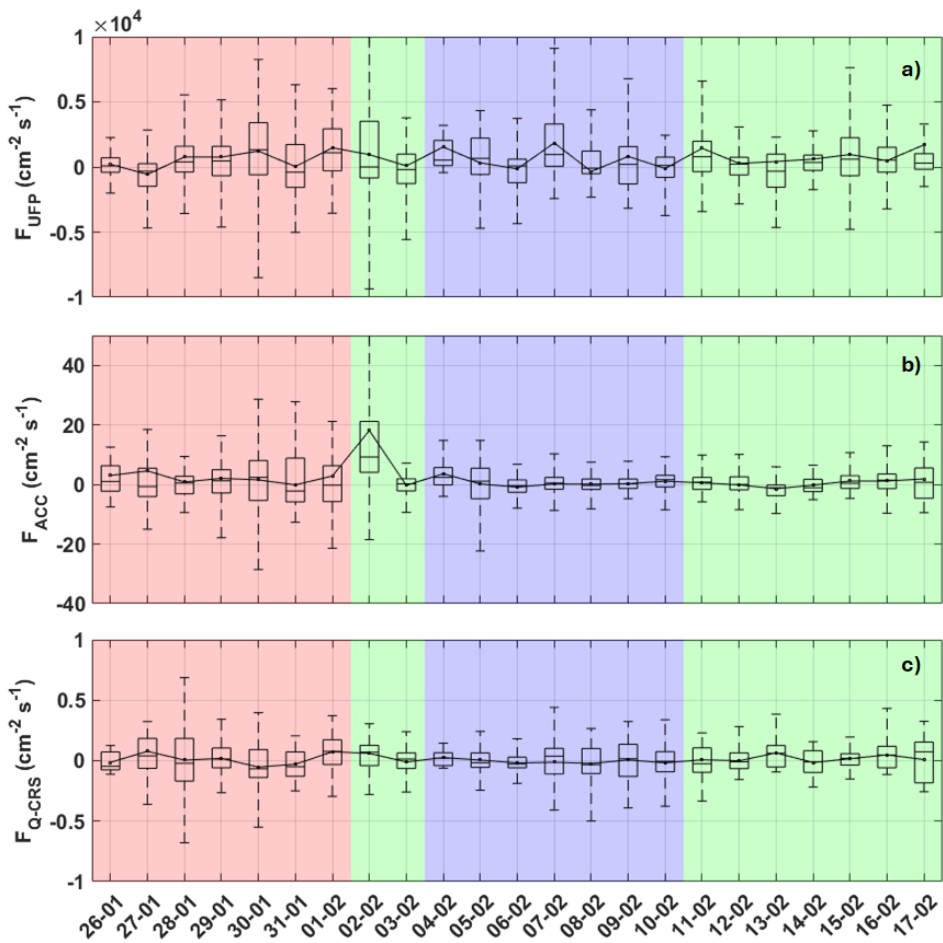

**Figure 5: Daily box plots for (a) $F_{UFP}$, (b) $F_{ACC}$ and (c) $F_{Q-CRS}$. Boxes represent the 25th and 75th percentiles. Whiskers correspond to $\pm 2.7\sigma$ and 99.3% data coverage. The colour bands indicate the anticyclonic period AC (red), the cyclonic period C (blue) and the two transition periods T (green). The line inside each box indicates the median, while the dots represent the mean values. The percentage of valid data is 67% during the anticyclonic period and 74% during the cyclonic period**

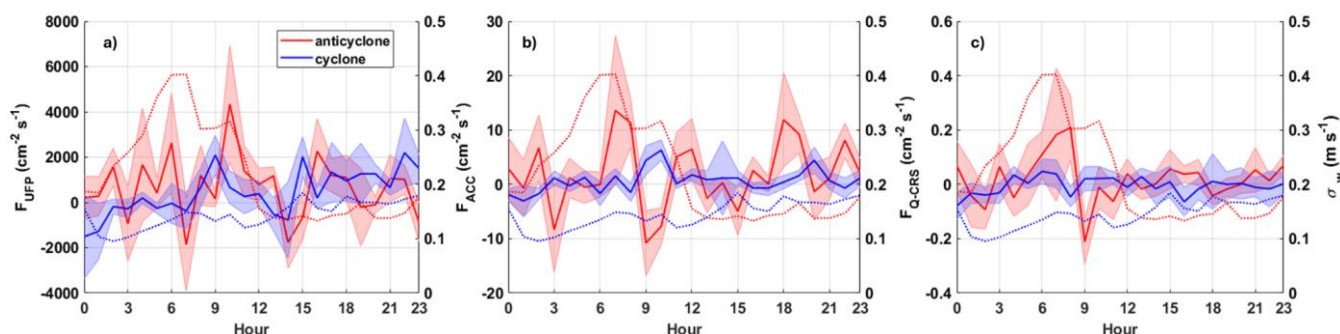

**Figure 6: Diurnal cycles (mean) of particle flux (net values, including both positive and negative fluxes; left axis) and standard deviation of vertical wind speed ($\sigma_w$; right axis for each panel - dashed line) at UAF-Farm for a) $F_{UFP}$, b) $F_{ACC}$ and c) $F_{Q\text{-}CRS}$. The diurnal trends are shown both for the anticyclonic and cyclonic period. Continuous lines are the mean value. The shadow area represents the standard error. Hour is in standard local time (LT).**

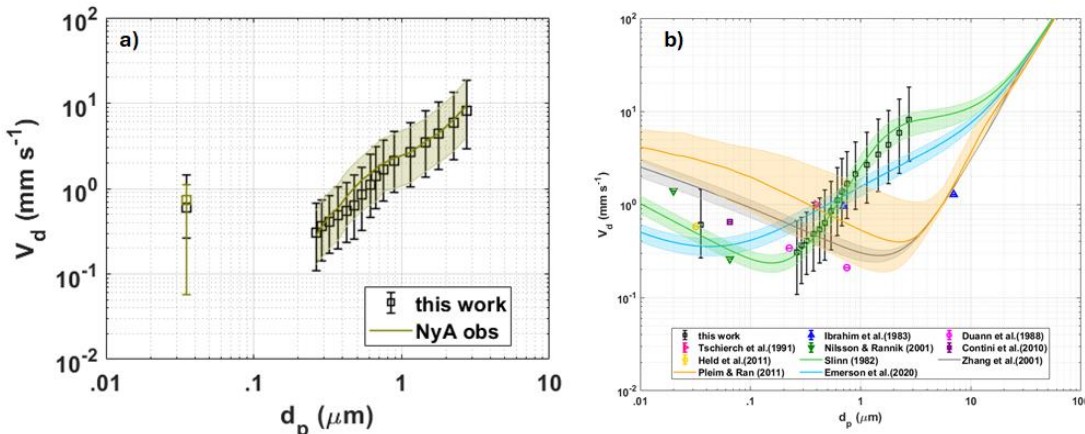

**Figure 7: (a) Median $V_d$ as a function of the geometric mean diameter measured during the campaign. Median values measured $V_d$ are compared to previous measurements of the deposition velocities on snow at Ny-Ålesund (Donateo et al., 2023). (b) A comparison with the model predictions for $V_d$. Observations are shown in symbols and models in lines. Error bars and shaded areas represent the interquartile range.**

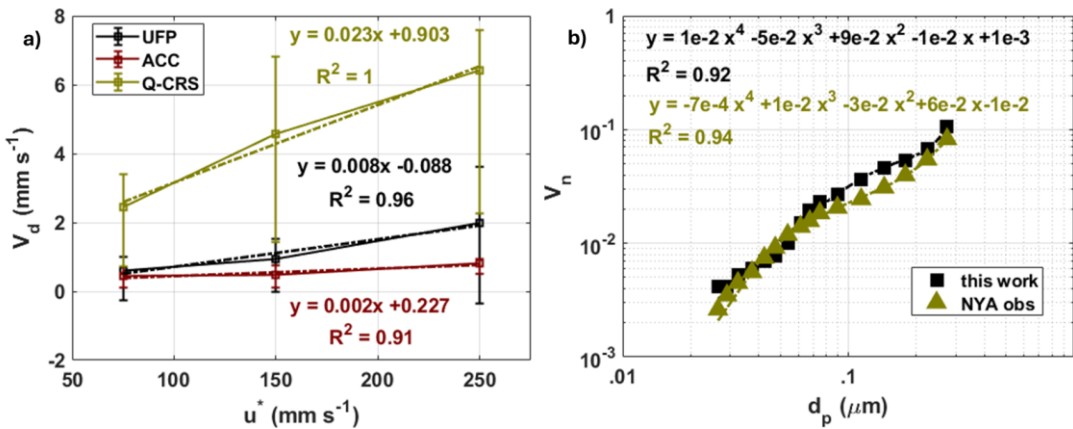

**Figure 8: (a)** Relation between median $V_d$ and u* for the different size ranges. Vertical bars represent the interquartile range for $V_d$ within the specific interval of u*. Friction velocity intervals were selected to optimise the number of data points within each interval and, hence, provide a statistically reliable median $V_d$. **(b)** Functional fit for $V_n$ as a function of the geometric mean particle diameter (excluded UFP) for this work observations and Ny-Ålesund ones.

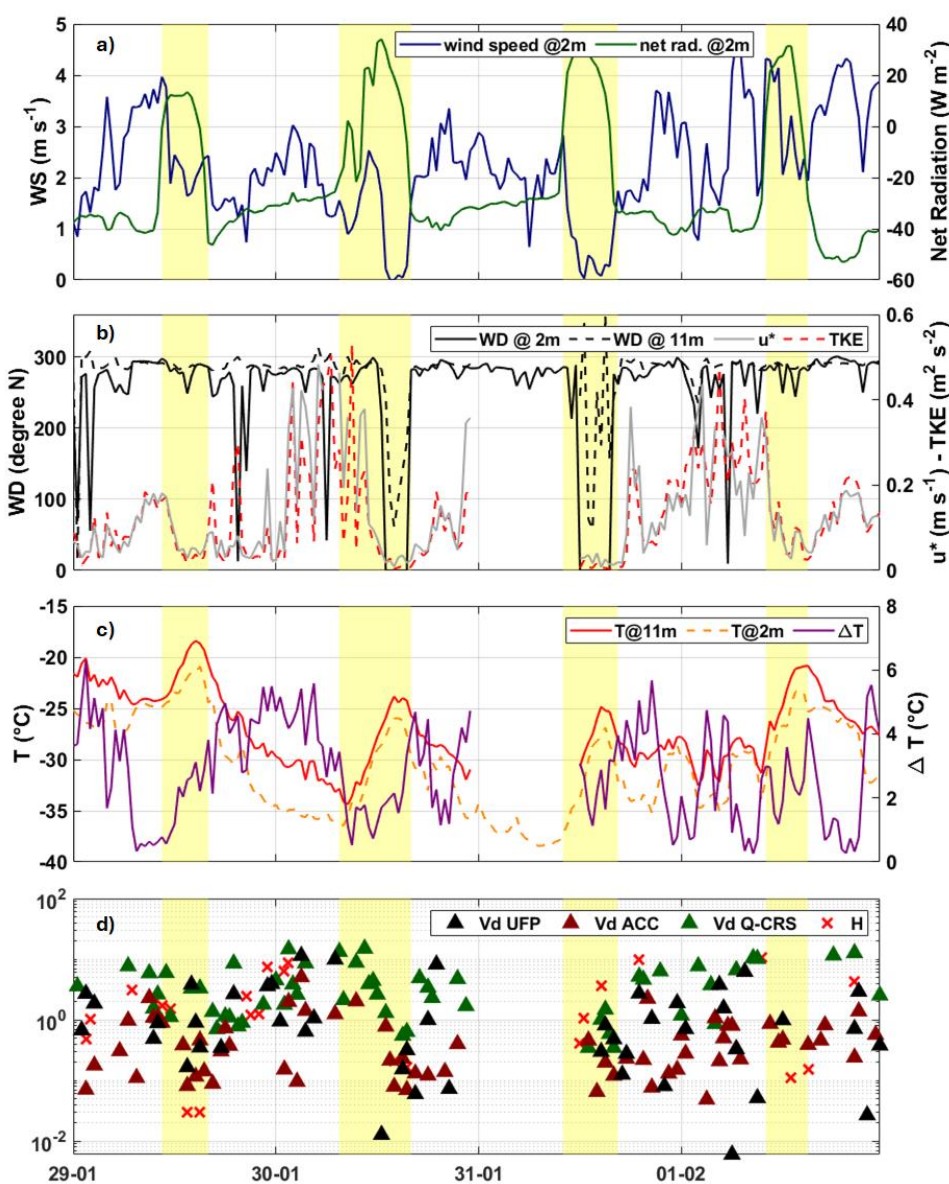

**Figure 9: Evolution of meteorological parameters and surface fluxes under typical anticyclonic conditions (from 29th January to 2nd February). The colour bars indicate AC$_a$ (no colour) and AC$_b$ (yellow) regimes. Notice that sensible heat (H) and particle V$_d$ are in log scale. Only positive (upward) H fluxes are considered in this analysis. Panel d) shows both deposition velocity (V$_d$, mm s$^{-1}$) and sensible heat flux (H, W m$^{-2}$) on the same y-axis; units for each variable are given here and can also be found in the main text.**

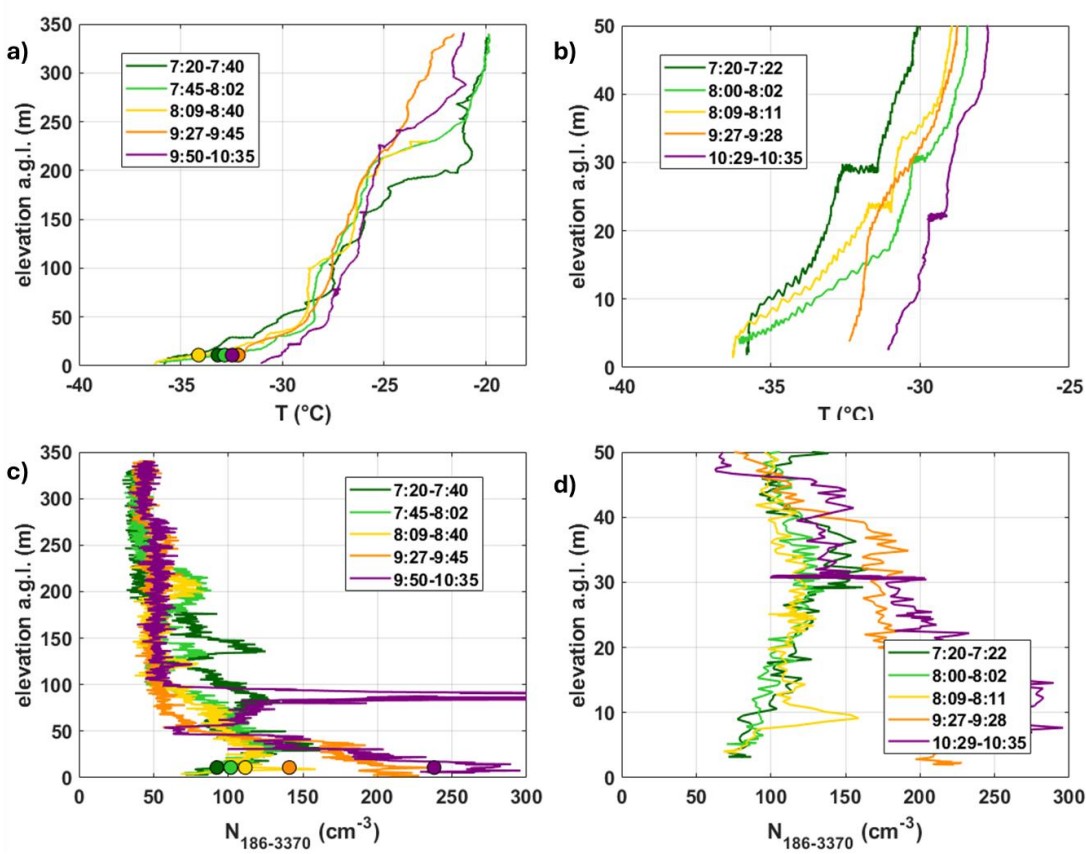

**Figure 10: Vertical profiles of temperature (upper panels) and accumulation- and coarse-mode aerosols (N$_{186\text{-}3370}$) (below panels) along five helikite profilings on 30$^{th}$ January 2022 between 07:20 and 10:35 LT. In (a) and (c) the EC measurements at 11 m are reported as coloured circles overlapping the helikite profiles. Time spent with the helikite hovering is sometimes marked by temperature variations at constant elevation. In (b) and (d) a zoom of the graph in (a) and (c) panels is reported.**

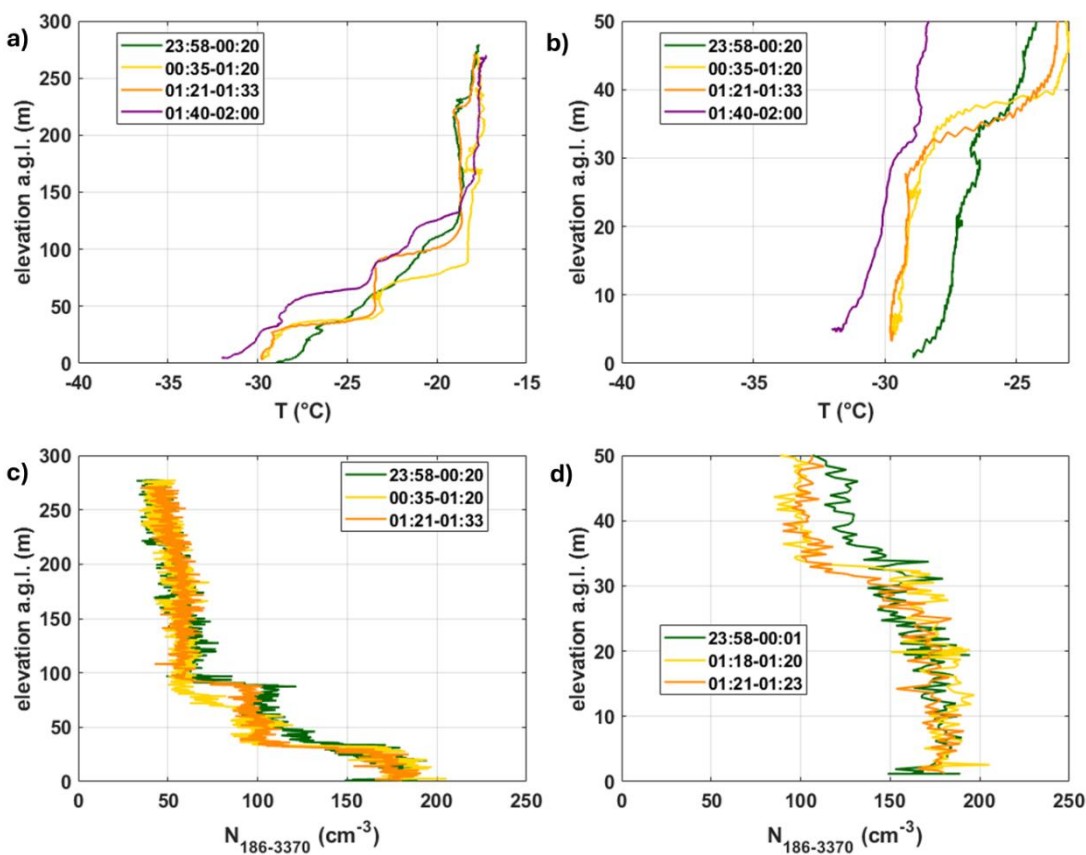

**Figure 11: Vertical profiles of temperature (upper panels) and accumulation- and coarse- mode aerosols ($N_{186-3370}$) (below panels) along, respectively, four and three helikite profilings between 31st January 23:58 LT and 1st February 2022 01:33 LT. In this case, on 31st January, EC measurements were interrupted for a technical issue. In (b) and (d) a zoom of the graph in (a) and (c) panels is reported.**

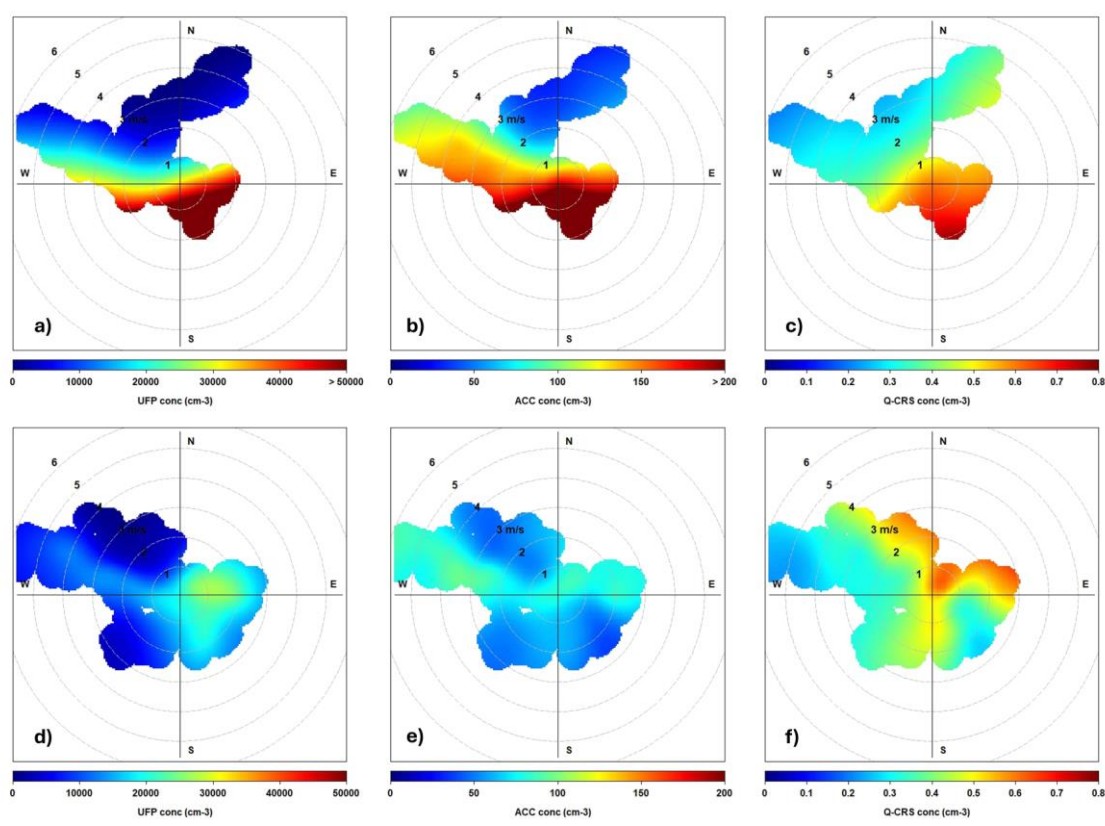

**Fig. A1. Bivariate polar plots (pollution roses) relating (a) UFP, (b) ACC and (c) Q-CRS particles mode in the anticyclonic (upper panels) and cyclonic (bottom panels) period.**

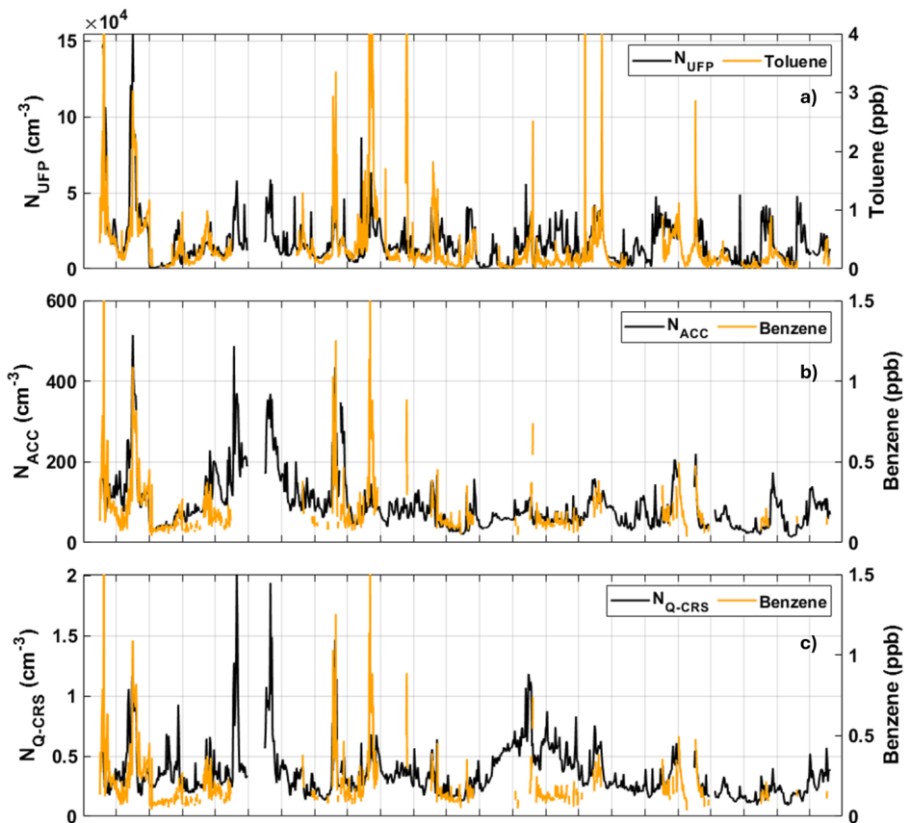

**Fig. A2.** Time series of the number particle concentration for (a) UFP, (b) ACC and (c) Q-CRS size range. In (a) on the right axis the toluene concentration (in ppb) is reported. In (b) and (c) is reported, on the right axis, the benzene concentration (in ppb)

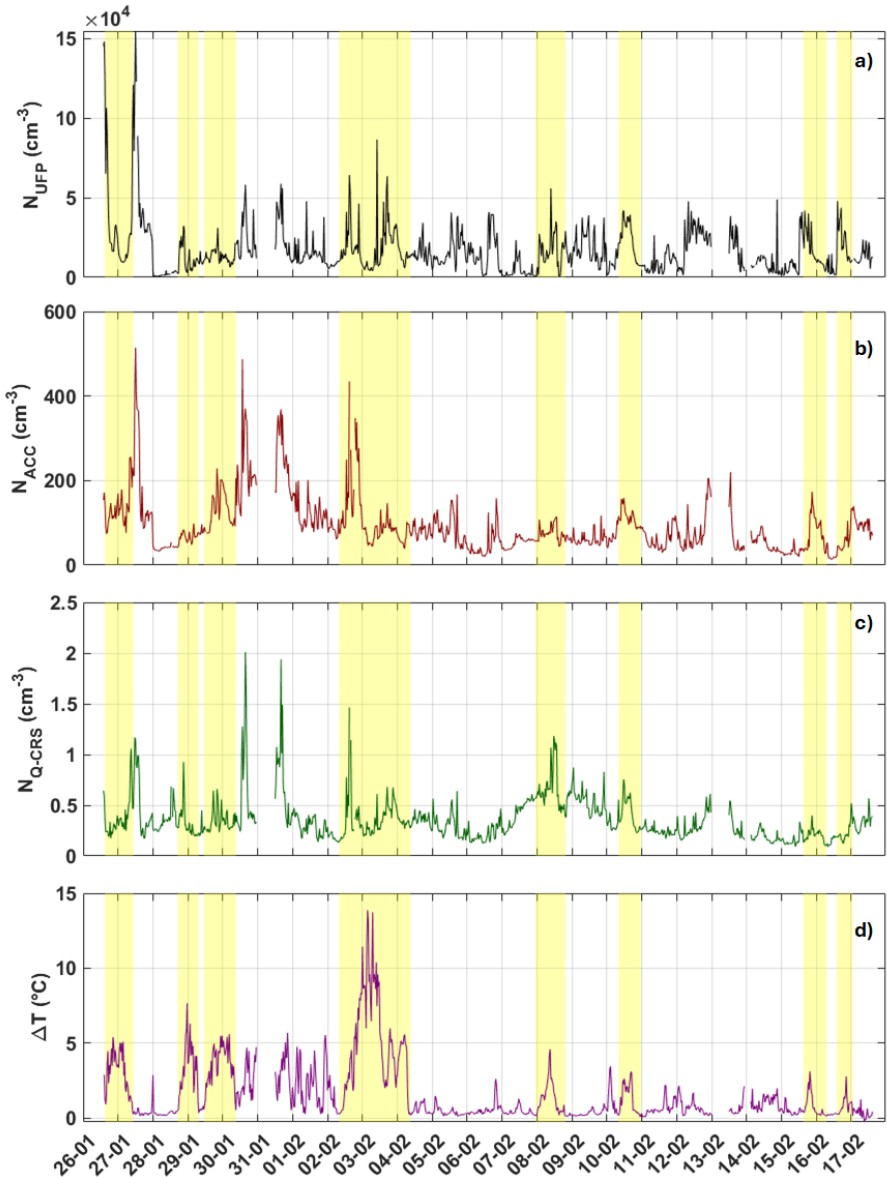

**Figure A3: Time series for (a) N$_{UFP}$, (b) N$_{ACC}$ and (c) N$_{Q\text{-}CRS}$. (d) Time series for ΔT during the measurement campaign. The yellow bands indicate a relevant value for ΔT.**

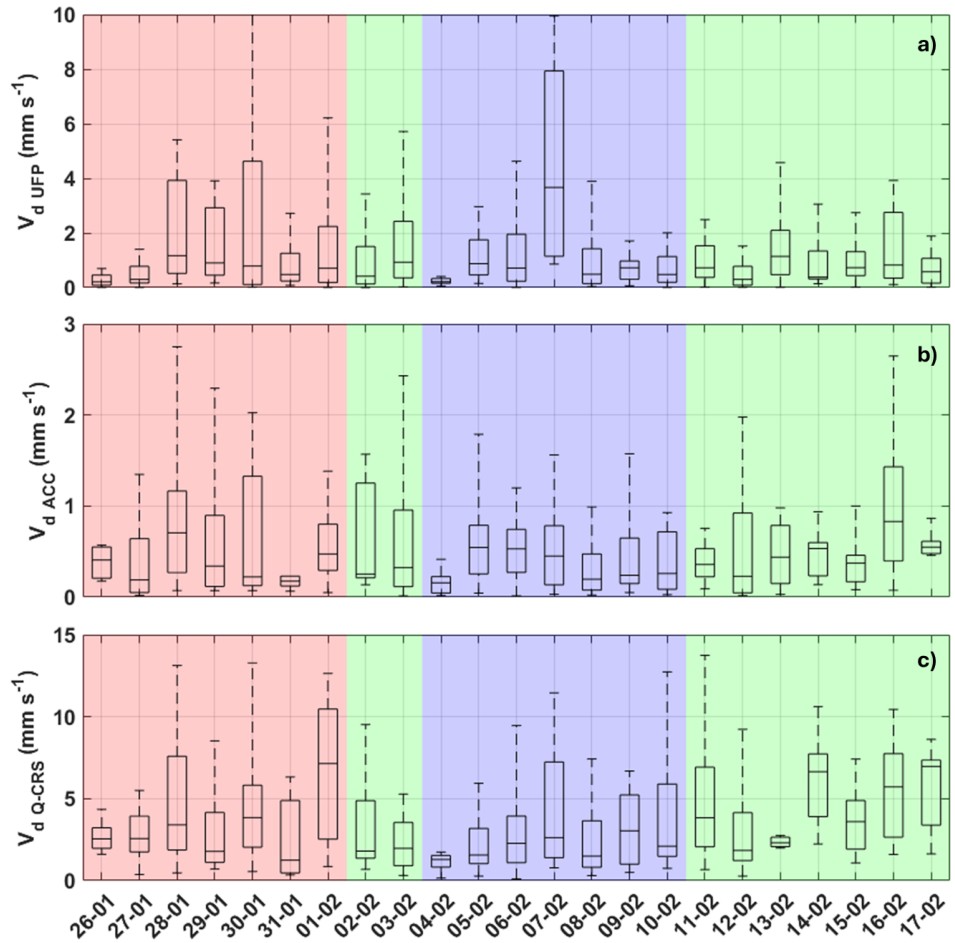

**Figure A4: Daily box plots for (a) Vd$_{UFP}$, (b) Vd$_{ACC}$ and (c) Vd$_{Q-CRS}$. Boxes represent the 25th and 75th percentiles. Whiskers correspond to ± 2.7σ and 99.3% data coverage. The colour bands indicate the anticyclonic period AC (red), the cyclonic period C (blue) and the two transition periods T (green). The line inside each box indicates the median.**

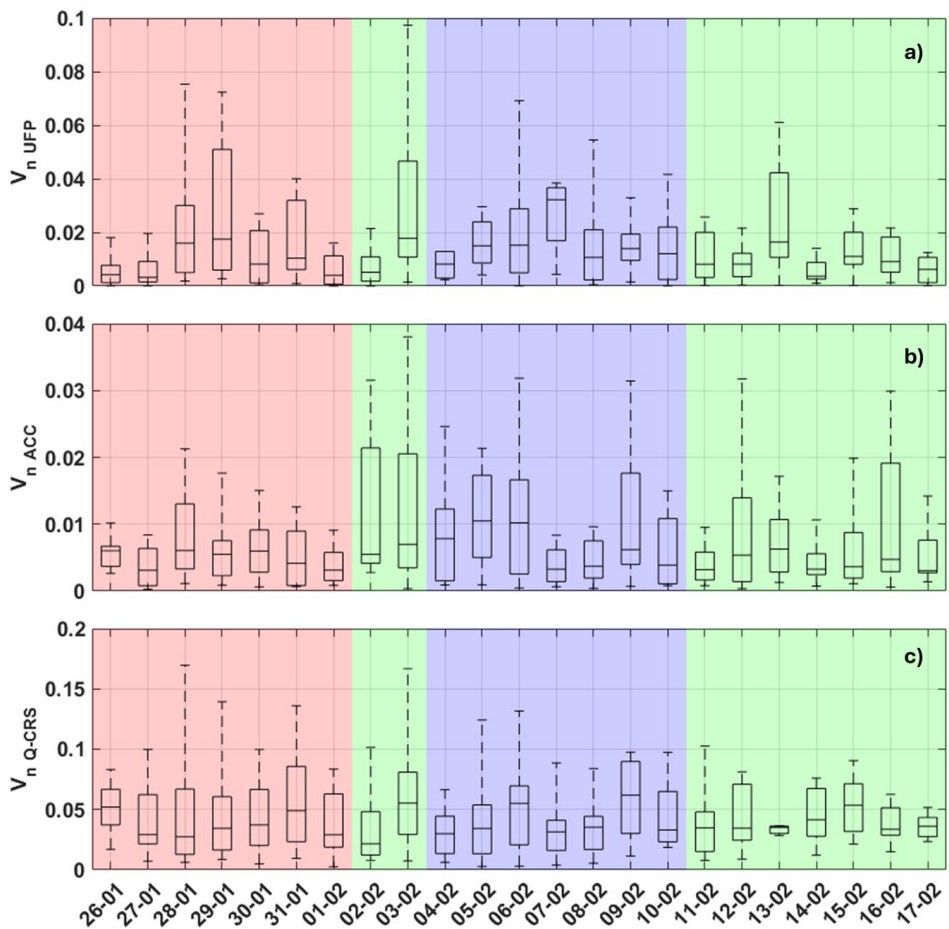

**Figure A5: Daily box plots for (a) Vn_UFP, (b) Vn_ACC and (c) Vn_Q-CRS. Boxes represent the 25th and 75th percentiles. Whiskers correspond to ± 2.7σ and 99.3% data coverage. The colour bands indicate the anticyclonic period AC (red), the cyclonic period C (blue) and the two transition periods T (green).  The line inside each box indicates the median.**

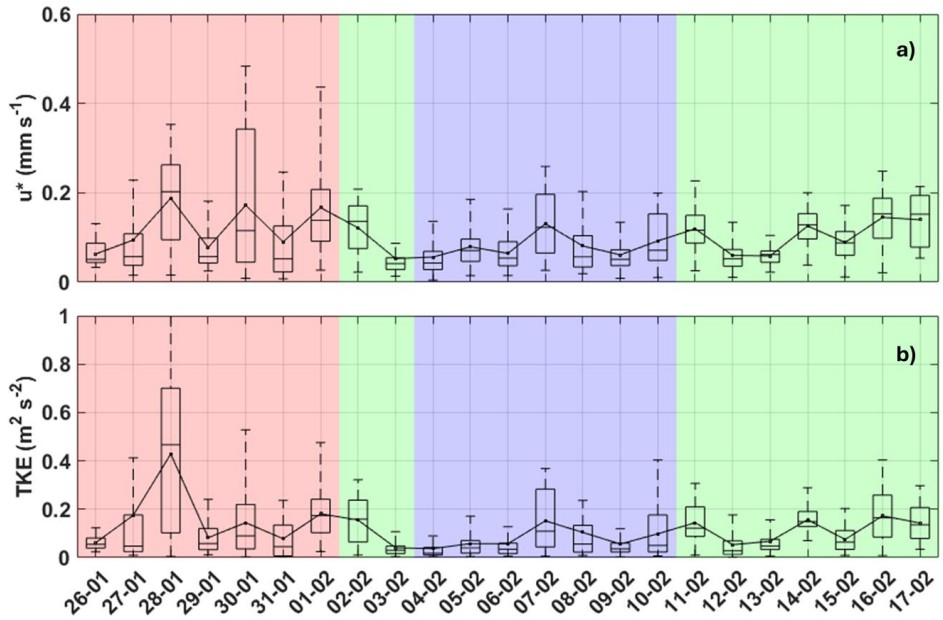

**Figure A6: Daily box plots for (a) friction velocity u\*, and (b) turbulent kinetic energy TKE. Boxes represent the 25th and 75th percentiles. Whiskers correspond to ± 2.7σ and 99.3% data coverage. The colour bands indicate the anticyclonic period AC (red), the cyclonic period C (blue) and the two transition periods T (green). The line inside each box indicates the median, while the dots represent the mean values.**