# Peer review of "Aerosol dry deposition fluxes on snow during the ALPACA campaign in Fairbanks, Alaska"

_EGUsphere, 2025_

## Referee Comment (RC1)

**Review: "Aerosol dry deposition fluxes on snow during the ALPACA campaign in Fairbanks, Alaska"**
Donateo et al.

This paper reports on calculated modal and size-resolved deposition fluxes from 2-months of surface measurements at a site in the urban Arctic (Fairbanks, Alaska). They find an increase in deposition flux with increasing diameter, consistent with previous work in a similar Arctic region and a model parameterization. Using temporal trends, meteorological regime classifications, and profiles of atmospheric thermodynamics and number concentrations, the authors attribute the variability in deposition fluxes to regional sources (anthropogenic pollution) with enhancements due to local meteorology and synoptic circulation.

This work provides very interesting insights on aerosol deposition in an understudied region that is strongly impacted by intraseasonal synoptic variability and distinct aerosol sources and transport. The discussions and presentation of results are prepared very well, and I am overall confident in the fidelity of this work. I can recommend this paper be considered for publication after the following minor comments, clarifications, and questions are addressed.

**Comments:**
- Lines 195-196: Can the authors please specify how the ultrafine particle mode was determined? Was this a subtraction of the OPC integrated number concentration from the total concentration measured by the CPC?

- Lines 283-287: Particle number concentrations in the tens of thousands seem exceptionally high for ambient measurements, even during Arctic hazy periods. It makes sense that the UFP mode would be the highest particle number concentration, but the value seems dramatically high. Can the authors please provide context for these number concentrations? First, as a comparison to previous measurements of total and modal number concentrations at Fairbanks/Urban Arctic and second with respect to the meteorology. Is there a sense of how high/different these aerosol number concentrations are from "background" conditions in Fairbanks or nearby/similar regions? Were the aerosol concentrations (total and modal) different between meteorological regimes? Additionally, typically in CPCs typically the laser becomes saturated above 10,000 $cm^{-3}$ and coincidence errors become quite large. Can the authors speak to and justify why they feel such high concentrations measured by the CPC can be trusted?

- Table 1: is the mean diameter (dp) in this table taken as the average of the range of diameters in each mode/size range? I believe this should be specified somewhere in the text and table legend as not to confuse with a geometric mean diameter from a modal fit to a size distribution.

- Did the authors test for independence in the 3 modes considered? More specifically, did they find coverability between any of the modes and does that have any influence on the interpretation of sources and deposition?

- I believe it is appropriate to provide standard deviations when means are reported.

- Section 3.2: Figures 3 and A1 should be cited in lines 283-287 (preferably with Table 1) before their first citation later in this section so the reader can see the trends and ranges discussed.

- Lines 288-301: Can the authors please quantify the particle differences in this passage. Phrases like "enhanced pollution", "concentration significantly smaller", "close to levels" and reference to the figure panels are used, but it helps the reader glean differences when these values are quantified. Did the authors also consider performing statistical tests on the differences between number concentrations in different meteorological regimes?

- Lines 300-301: There is higher $N_{Q-CRS}$ in periods before Feb 7-10 (gleaning from Fig. 3 and Fig. A1), but the authors state that a maximum is observed during Feb 7-10. Do the authors mean elevated number concentrations during that period? Please clarify. Additionally, the authors should clarify what is meant by "… is characteristic of this class of particles and may witness a contribution from regional transport (e.g. dust or sea salt particles)." I believe what the authors are saying here is that dust and sea salt are primarily in the coarse mode and an elevation in number concentration of these particles may suggest an increase in their transport. This should be clearly stated and with citations supporting these claims (1 - dust/sea salt in the coarse mode, 2 – elevation in number concentration from transport).

- Section 3.3: The authors should place the calculated values in context with previous measurements. How do these flux values compare to other Arctic or continental regions? A diel cycle of modal flux is an interesting result that needs to be compared to similar and dissimilar calculations.

- Lines 336-337: Please state the quantified difference between FACC during ACC compared to the other periods that is referenced here.

- Figure 3 and 4: Please denote that (if I assume correctly) dots in the box plots represent the mean and line in the box is the median.

- Lines 358-361: The authors suggest wind-driven resuspension as a potential mechanism contributing to the variability in FQ-CRS. Did the authors observe more variable FQ-CRS with variable or stronger wind speeds?

- Lines 360-361: The authors spend little time discussing the differences in the flux diurnal cycle between synoptic regimes. I think the larger fluxes throughout the day

during the anticyclonic regime is consistent with the overall finding of higher number concentrations and higher fluxes during the anticyclonic regime and should be stated.

- Line 384-385: Citation to previous literature should be provided here for this claim.

- Line 397-400: Are the deposition velocities cited from previous measurements in these lines from Arctic/polar regions? Please clarify.

- Line 401-402: Are the authors able to correlate the parameterized deposition velocities using Slinn (1982) with the calculated median values shown in Fig. 5b from this study? How well do they agree? Further, are the author able to speculate on why the Slinn parameterization has such (visually) great agreement with your calculated values from these measurements?

- Figure 8 caption: it should be clarified that $V_n$ is the deposition velocity normalized by the friction velocity.

- Line 435: I might suggest referring to the 0.54-0.89 µm size range as the "large" accumulation mode here.

- Line 438: I understand the suggestion that physical characteristics (like different particle densities) may affect the deposition behavior in this mode, but can the authors clarify why hygroscopitiy would affect the deposition behavior? A citation might be helpful here.

- Lines 438-441: Citations are needed here to allude to these suggested effects.

- Fig 9: The authors should provide units for the y-axis in panel d).

- Fig 10 caption: It should be specified that panels (b,d) are zoomed in versions of panel (a,c).

---

## Referee Comment (RC2)

This study looked at how aerosols exchange near the surface in a cold, snowy Arctic city, Fairbanks, Alaska. The authors used EC during a 23-day period in winter 2022 to measure whether particles were rising into the air (emitted) or falling back to the ground (deposited). They studied different particle sizes, including ultrafine particles, mid-sized particles, and larger ones that are more likely to settle.

They found that the smallest particles were mostly emitted, especially during the day, likely due to traffic and other local sources. They also found that the bigger particles behaved differently. They were often depositing to the ground, especially when the weather was calm and cold (AC). But these upward and downward movements didn't always match up with the particle concentration in the air, showing that just looking at concentration isn't enough to understand pollution. Importantly, they discovered that synoptic conditions played a major role. When the wind was weak and the air was still, pollution built up near the ground. But stronger winds and temperature differences between the air and the snowy ground helped push particles back down, even if there weren't many in the air.

A major finding was that downward movement (deposition) didn't just happen because of gravity. Instead, turbulence in the air was a key driver. In extremely cold conditions, sharp temperature differences near the snowy surface may have created an extra push for particles to settle, a process known as thermophoresis. This may be a missing piece in current models, potentially leading to an underestimation of dry deposition rates.

Overall, the manuscript is well structured and scientifically sound. I have some comments that I believe, once addressed, will further strengthen the clarity and impact of the study and bring it to a publishable standard.

**Major comment:**

The manuscript discusses the likely influence of local sources (e.g., traffic, UAF activity) and regional sources (e.g., downtown Fairbanks) based primarily on the diurnal patterns. While these observations are suggestive, they remain qualitative. To strengthen the attribution of aerosol sources under different synoptic regimes, I suggest incorporating back trajectory analyses (e.g., using HYSPLIT or similar tools). Stratifying trajectories by synoptic condition and/or flux event (e.g., high UFP emissions or deposition episodes) would provide more robust evidence for the origin of aerosols and help distinguish between local vs. advected pollution events. This would also complement the existing discussion of transport under anticyclonic vs cyclonic regimes. If high-resolution meteorological data to run the back trajectory analysis is not available to resolve local-scale features, extending the footprint-based approach separately under different synoptic conditions would still provide a valuable, site-relevant alternative. A brief discussion of typical source characteristics by particle size (e.g., UFP, ACC, and Q-CRS) would also be helpful to support this analysis.

Throughout the manuscript, inversion layers and stability regimes and vertical profile of atmosphere are discussed as key factors influencing aerosol fluxes. However, the meteorological context is not sufficiently demonstrated. I suggest including synoptic-scale information, such as MSLP composites, and vertical profiles (e.g., radiosonde soundings), to

better illustrate the evolution and structure of inversion layers during the campaign periods for different dominant synoptic conditions.

**Minor comments:**

The manuscript does not discuss whether EC has previously been used in snow-covered or Arctic urban environments, nor does it acknowledge any potential limitations of using this technique under such conditions. If this study is among the first to apply EC to aerosol deposition in Arctic snow-covered urban areas, it would be helpful to state this clearly and reflect on associated uncertainties.

Figure 1 shows the spatial distribution of the flux footprint, but it lacks clarification of what the coloured contours represent.

In Section 2.1, you mention several nearby potential aerosol sources, including the Tanana River. To support later discussions on aerosol sources under different synoptic conditions, I suggest highlighting these key features (e.g., the river, airport, major roads) directly to Figure 1. This would provide helpful spatial context when interpreting the footprint and probably back trajectories influences on observed fluxes.

Please clarify whether a constant $z = 11$ m (instrument height) was used for the calculation of the atmospheric stability parameter $\zeta = z/L$. Also, briefly describe how the Obukhov length $L$ was derived.

Regarding the data processing and all filtration processes described in lines 171 to 202, clarify on how representative the final dataset is. Especially whether there is sufficient coverage across stability regimes, times of day, or meteorological conditions. Also in line 171, you mention that data were discarded for being "outside the absolute limits." Could you clarify what specific thresholds were used in this step?

Please clarify on the average error of 101% in line 193.

The final sentence on page 16 states that particle fluxes during the cyclonic period are "on average null for the whole day" (referring to Fig. 6). However, this statement does not appear to be consistent with panel (a) of Fig. 6 - for UFPs. Please clarify or revise this statement for accuracy.

In the conclusion of Section 4.2, it has been stated that the UAF Farm site is devoid of local sources and that aerosol concentrations are mainly influenced by transport. However, earlier in the manuscript (e.g., Section 2.1), the potential influence of local sources such as the nearby airport has been discussed. Please clarify what is meant by "local" in this context and reconcile these two statements to avoid confusion.

In lines 306–314, the elevated night time accumulation-mode particle fluxes are attributed to re-entrainment under stable, calm conditions. While this is a plausible mechanism, the role of varying emissions, such as increased night time residential heating or episodic sources is not discussed. How did you ignore considering whether higher night time emissions may also

contribute to the observed fluxes, particularly since these could be source-driven rather than solely due to surface-layer turbulence.

Since the mean values are available in the boxplots in Figure 3 & 5 (I assume the dots are the mean values), Table 1 may no longer be necessary (maybe keeping it in the supplementary materials). Presenting this information visually would improve clarity and reduce redundancy.

In Figure 6, the diurnal variability in emission fluxes is interpreted primarily as a result of traffic emissions, based on the timing of observed peaks. While this is plausible, I recommend also considering the role of deposition (i.e., net negative fluxes) and turbulence strength (e.g., vertical wind speed or shear) in shaping the diurnal pattern. Do they show similar consistency? Including these factors would provide a more complete picture of the boundary-layer processes controlling flux variability, and help to better separate source-driven vs. transport/mixing influences.

In Section 3.4 and Table 2, the transfer velocity $v=F/N$ is only presented as a mean value. For consistency with Figures 3 and 5, and to better capture variability and uncertainty, why not presenting them as a boxplot grouped by stability class and particle size. This would allow readers to more directly compare patterns in $v$ with those in $N$ and $F$, and assess whether the observed means are representative of the distribution.

Why haven't you had included time series plots of $V_n$, $V_d$, and friction velocity ($u^*$)? This would help the reader assess how these variables evolve and interact dynamically, especially under changing meteorological conditions.

Where is equation 3 mentioned in line 429?

In Section 3.6, the discussion of meteorological and deposition differences across regimes (e.g., ACa, ACb, C, T1, T2) is presented primarily in qualitative terms (e.g., "median u* was higher...", "Vd values were greater..."). I suggest making this section more quantitatively grounded by explicitly referencing values from Tables 4 and 5 within the text to support these comparisons. Additionally, to assess whether the observed differences are statistically meaningful, I recommend performing basic statistical tests on key variables such as deposition velocity or ΔT. This would help substantiate the conclusions about the influence of meteorological regimes on aerosol deposition and strengthen the interpretation of the results.

The observation that median deposition velocities for ACC particles are lower than for UFPs is counterintuitive given their larger size. This might reflect differences in turbulence coupling, rebound effects, or diffusional deposition mechanisms, but it would be helpful if the authors could provide a brief discussion or hypothesis to interpret this behavior.

While the surface-based eddy covariance measurements provide valuable insight into particle emission and deposition processes, the study would benefit from acknowledging the limitation posed by the lack of aerosol flux measurements at the top of the boundary layer. Without data on vertical flux divergence, it is difficult to fully constrain the aerosol budget or to distinguish surface-driven processes from entrainment or dilution effects. A brief

discussion of this limitation, and its implications for interpreting the observed surface fluxes, would improve the overall clarity and transparency of the analysis.

---

## Author Comment (AC1)

**Article Discussion comments for: "Aerosol dry deposition fluxes on snow during the ALPACA campaign in Fairbanks, Alaska", Donateo et al., Atmospheric Chemistry and Physics, Manuscript ID: egusphere-2025-1366**

We thank the Reviewer for his/her careful assessment of our manuscript and his/her valuable suggestions. We found them very useful for improving and clarifying some of the information that was unclear in the submitted manuscript. Below, we respond to the comments in turn, and summarise modifications made to the manuscript. Our responses are formatted as follows:

Reviewer comments **- black text (bold)**

Author responses - black text

Revised manuscript text - green text

Line numbers refer to those in the original submission.

**Reviewer #1:** RC1: "Comment on egusphere-2025-1366' - https://doi.org/10.5194/egusphere-2025-1366-RC1

**This paper reports on calculated modal and size-resolved deposition fluxes from 2-months of surface measurements at a site in the urban Arctic (Fairbanks, Alaska). They find an increase in deposition flux with increasing diameter, consistent with previous work in a similar Arctic region and a model parameterization. Using temporal trends, meteorological regime classifications, and profiles of atmospheric thermodynamics and number concentrations, the authors attribute the variability in deposition fluxes to regional sources (anthropogenic pollution) with enhancements due to local meteorology and synoptic circulation.**

**This work provides very interesting insights on aerosol deposition in an understudied region that is strongly impacted by intraseasonal synoptic variability and distinct aerosol sources and transport. The discussions and presentation of results are prepared very well, and I am overall confident in the fidelity of this work. I can recommend this paper be considered for publication after the following minor comments, clarifications, and questions are addressed.**

**Comments:**

**• Lines 195-196: Can the authors please specify how the ultrafine particle mode was determined? Was this a subtraction of the OPC integrated number concentration from the total concentration measured by the CPC?**

In the new version of the manuscript a sentence has been added to clarify how the ultrafine (UFP) number concentration has been calculated.

Ultrafine particle concentration (UFP) was obtained as the difference between the total number concentration (CPC measurement) and the OPC integrated concentration in the size range 0.25–1 μm.

**• Lines 283-287: Particle number concentrations in the tens of thousands seem exceptionally high for ambient measurements, even during Arctic hazy periods. It makes sense that the UFP mode would be the highest particle number concentration, but the value seems dramatically high. Can the authors please provide context for these number concentrations? First, as a comparison to previous measurements of total and modal number concentrations at Fairbanks/Urban Arctic and second with respect to the meteorology.**

We appreciate the reviewer's concern and agree that providing context is important. We have therefore added a short comparison with previous studies conducted in Fairbanks in Sect. 3.2.

The particle number concentrations observed in this study are consistent with previous measurements reported for Fairbanks during the winter season. For example, Robinson et al. (2023) documented a median particle number concentration above $4.5 \times 10^4$ cm$^{-3}$ during cold stagnation events, with UFPs accounting for most particles (> 95%). Again, Robinson et al. (2023), measured the highest UFP number concentration (7.2 x $10^4$ cm$^{-3}$) in Downtown East (Fairbanks).

Robinson, E. S., Cesler-Maloney, M., Tan, X., Mao, J., Simpson, W., and DeCarlo, P. F.: Wintertime spatial patterns of particulate matter in Fairbanks, AK during ALPACA 2022, Environm. Sci. Atmospheres, 3, 568–580, https://doi.org/10.1039/D2EA00140C, 2023.

**Is there a sense of how high/different these aerosol number concentrations are from "background" conditions in Fairbanks or nearby/similar regions?**

We agree that it is important to contextualize our observations with respect to background conditions. We have therefore added to the revised version of the manuscript a short discussion comparing the

particle number concentrations observed during our campaign with background levels reported in Fairbanks and in other Arctic remote locations.

The particle number concentrations observed in this study in the immediate outskirts of Fairbanks are comparable to, or slightly higher than, those previously reported in the surroundings of Fairbanks. For instance, Robinson et al. (2023) measured concentrations on the order of $1.5 \times 10^4$ cm$^{-3}$ at sites located on the hills north of the city during strong inversion conditions. By contrast, typical particle number concentrations at pristine Arctic sites, such as Barrow in Alaska (Rose et al., 2021) or Zeppelin observatory in Ny-Ålesund (Croft et al. 2016) are two to three orders of magnitude lower ($10^2$–$10^3$ cm$^{-3}$), underscoring the dominant impact of local sources and boundary-layer processes in shaping aerosol levels in Fairbanks. Our observations thus align with pollution episodes previously described for the region and highlight the strong contrast between clean background conditions and the highly elevated concentrations associated with persistent inversions and limited boundary-layer mixing.

Rose, C., Collaud Coen, M., Andrews, E., Lin, Y., Bossert, I., Lund Myhre, C., Tuch, T., Wiedensohler, A., Fiebig, M., Aalto, P., Alastuey, A., Alonso-Blanco, E., Andrade, M., Artíñano, B., Arsov, T., Baltensperger, U., Bastian, S., Bath, O., Beukes, J. P., Brem, B. T., Bukowiecki, N., Casquero-Vera, J. A., Conil, S., Eleftheriadis, K., Favez, O., Flentje, H., Gini, M. I., Gómez-Moreno, F. J., Gysel-Beer, M., Hallar, A. G., Kalapov, I., Kalivitis, N., Kasper-Giebl, A., Keywood, M., Kim, J. E., Kim, S.-W., Kristensson, A., Kulmala, M., Lihavainen, H., Lin, N.-H., Lyamani, H., Marinoni, A., Martins Dos Santos, S., Mayol-Bracero, O. L., Meinhardt, F., Merkel, M., Metzger, J.-M., Mihalopoulos, N., Ondracek, J., Pandolfi, M., Pérez, N., Petäjä, T., Petit, J.-E., Picard, D., Pichon, J.-M., Pont, V., Putaud, J.-P., Reisen, F., Sellegri, K., Sharma, S., Schauer, G., Sheridan, P., Sherman, J. P., Schwerin, A., Sohmer, R., Sorribas, M., Sun, J., Tulet, P., Vakkari, V., van Zyl, P. G., Velarde, F., Villani, P., Vratolis, S., Wagner, Z., Wang, S.-H., Weinhold, K., Weller, R., Yela, M., Zdimal, V., and Laj, P.: Seasonality of the particle number concentration and size distribution: a global analysis retrieved from the network of Global Atmosphere Watch (GAW) near-surface observatories, Atmos. Chem. Phys., 21, 17185–17223, https://doi.org/10.5194/acp-21-17185-2021, 2021.

Croft, B., Martin, R. V., Leaitch, W. R., Tunved, P., Breider, T. J., D'Andrea, S. D., and Pierce, J. R.: Processes controlling the annual cycle of Arctic aerosol number and size distributions, Atmos. Chem. Phys., 16, 3665–3682, https://doi.org/10.5194/acp-16-3665-2016, 2016

**Were the aerosol concentrations (total and modal) different between meteorological regimes?**

Yes, this aspect is addressed in the manuscript. In Sect. 3.6 we specifically discuss the variability of aerosol concentrations, fluxes, and deposition velocities ($V_d$) under different meteorological regimes. Tables 4 and 5 report the corresponding concentration values and flux characteristics for the main synoptic categories (cyclonic, anticyclonic, etc.), showing that concentrations and fluxes indeed vary

depending on the prevailing regime. A sentence has been added at the end of Sect. 3.2 to reiterate more clearly these differences.

The highest concentrations occurred during persistent anticyclonic periods with strong surface-based inversions, weak winds, and low mixing heights, which favour the accumulation of locally emitted particles. By contrast, during frontal passages and enhanced mixing, number concentrations dropped by an order of magnitude. These results therefore fall within the expected range for Fairbanks wintertime conditions and reflect the strong modulation of aerosol concentrations by meteorology.

**Additionally, typically in CPCs the laser becomes saturated above 10,000 cm-3 and coincidence errors become quite large. Can the authors speak to and justify why they feel such high concentrations measured by the CPC can be trusted?**

We acknowledge that coincidence errors and laser saturation can be a concern in CPC measurements. However, in our case the instrument used was a TSI 3756, whose specifications (manufacturer's datasheet - https://tsi.com/getmedia/cb4a10a6-3ae8-4cb0-bb5d-1d9dfc841ded/3756_A4_5002016_RevB_Web?ext=.pdf, last access 25/08/2025) report a maximum measurable concentration of $3.0 \times 10^5$ particles $cm^{-3}$ before coincidence errors exceed 10%. The highest concentrations observed in our dataset were on the order of $2$–$3 \times 10^4$ particles $cm^{-3}$, well below this upper threshold. We therefore consider the measurements to be reliable and not significantly affected by coincidence errors. Although the CPC 3756 has not been the explicit subject of dedicated publications, many of its key features (2.5 nm sensitivity, butanol capacity, and data rates up to 50 Hz) are direct implementations of the same technology used in the CPC 3776, which is well described in the literature (Takegawa et al. 2016).

Takegawa, N., Iida, K., Sakurai, H.: Modification and laboratory evaluation of a TSI ultrafine condensation particle counter (Model 3776) for airborne measurements. Aerosol Science and Technology, 51:2, 235-245, DOI:10.1080/02786826.2016.1261990

• **Table 1: is the mean diameter (dp) in this table taken as the average of the range of diameters in each mode/size range? I believe this should be specified somewhere in the text and table legend as not to confuse with a geometric mean diameter from a modal fit to a size distribution.**

In the revised version of the manuscript, we have replaced the previous values with the mean diameter ($d_P$) of each size range, and we have specified this clearly in the caption of the corresponding table to avoid any confusion with a geometric mean diameter derived from a modal fit.

$d_p$ represents the arithmetic mean diameter of each size range.

**• Did the authors test for independence in the 3 modes considered? More specifically, did they find coverability between any of the modes and does that have any influence on the interpretation of sources and deposition?**

We thank the reviewer for this question. To ensure that the three modes considered were statistically independent, we evaluated the correlation among the different size channels measured by the Optical Particle Counter (OPC). Channels showing correlations higher than 50% were merged into the same size class. This procedure minimized overlap between adjacent modes and reduced the risk of double-counting or misinterpreting correlated particle populations. As a result, the three modes presented in the manuscript can be considered sufficiently independent for the purpose of source attribution and deposition analysis. We have clarified this point in the revised Sect. 2.3.

To assess the independence of the particle size modes, we calculated Pearson correlation coefficients between number concentrations in adjacent OPC size channels, merging those with a correlation > 0.5. This approach reduces coverability between size classes and ensures that the reported modes represent distinct particle populations for interpretation of sources and deposition processes.

**• I believe it is appropriate to provide standard deviations when means are reported.**

In Table 3 we already report both the standard deviation and the standard error together with the mean values, to provide a comprehensive representation of the data variability and uncertainty.

**• Section 3.2: Figures 3 and A1 should be cited in lines 283-287 (preferably with Table 1) before their first citation later in this section so the reader can see the trends and ranges discussed.**

In the revised version of the manuscript a citation to Fig. 3 and (now) Fig. A3 was inserted early in Sect. 3.2.

**• Lines 288-301: Can the authors please quantify the particle differences in this passage. Phrases like "enhanced pollution", "concentration significantly smaller", "close to levels" and reference to the figure panels are used, but it helps the reader glean differences when these values are**

**quantified. Did the authors also consider performing statistical tests on the differences between number concentrations in different meteorological regimes?**

In the revised manuscript, we have added explicit percentage differences to quantify the particle differences previously described qualitatively with terms such as "enhanced pollution," "concentration significantly smaller," or "close to levels". In addition, we have explicitly referenced Tables 4 and 5, from which readers can extract the numerical values. These additions allow the reader to clearly assess the magnitude of particle variations under different conditions.

We have performed statistical significance tests (Kruskal–Wallis) on the differences in number concentrations between the various meteorological regimes. The results indicate that all differences are statistically significant ($p < 0.05$). These results have been added to Sect. 3.2.

Statistical analysis using the Kruskal–Wallis test confirms that number concentrations differ significantly among all meteorological regimes ($p < 0.05$), supporting the observed variations discussed above.

**• Lines 300-301: There is higher NQ-CRS in periods before Feb 7-10 (gleaning from Fig. 3 and Fig. A1), but the authors state that a maximum is observed during Feb 7-10. Do the authors mean elevated number concentrations during that period? Please clarify.**

Yes, the authors indeed refer to elevated number concentrations during the period of February 7–10. We have clarified the text in the revised manuscript to make this explicit, specifying that the "maximum" mentioned corresponds to higher particle number concentrations rather than absolute peak fluxes.

An increase in particle number concentration $N_{Q-CRS}$ (up to 0.61 cm$^{-3}$) was observed (Fig. 3) during the C period (7 - 10 February).

**Additionally, the authors should clarify what is meant by "… is characteristic of this class of particles and may witness a contribution from regional transport (e.g. dust or sea salt particles)." I believe what the authors are saying here is that dust and sea salt are primarily in the coarse mode and an elevation in the concentration of these particles may suggest an increase**

**in their transport. This should be clearly stated and with citations supporting these claims (1 - dust/sea salt in the coarse mode, 2 – elevation in number concentration from transport).**

We thank the reviewer for pointing out this ambiguity. We agree that our statement should be clarified. What we meant is that dust and sea salt particles typically occur in the coarse mode, and that an increase in coarse-mode number concentrations may therefore indicate a contribution from regional transport of these particles. We have revised the text accordingly and added supporting citations.

This behaviour is characteristic of coarse-mode particles, which are commonly associated with primary emissions such as mineral dust and sea salt (Seinfeld & Pandis, 2016). An increase in the number concentration of coarse particles may therefore be indicative of long-range or regional transport events, when enhanced advection can bring dust or sea-salt aerosols into the measurement area (Textor et al., 2006), as already observed in the Fairbanks area since the early 90s' (Shaw 1991a,b).

Seinfeld, J.H., Pandis, S.N.: Atmospheric Chemistry and Physics: From Air Pollution to Climate Change, Wiley, New Jersey, 2016

Shaw, G. E., Aerosol Chemical Components in Alaska Air Masses 1 Aged Pollution, Journal of Geophysical Research, 96, 22357-22368, 1991a.

Shaw, G. E., Aerosol Chemical Components in Alaska Air Masses 2 Sea Salt and Marine Products, Journal of Geophysical Research, 96, 22369-22372, 1991b.

Textor, C., Schulz, M., Guibert, S., Kinne, S., Balkanski, Y., Bauer, S., Berntsen, T., Berglen, T., Boucher, O., Chin, M., Dentener, F., Diehl, T., Easter, R., Feichter, H., Fillmore, D., Ghan, S., Ginoux, P., Gong, S., Grini, A., Hendricks, J., Horowitz, L., Huang, P., Isaksen, I., Iversen, I., Kloster, S., Koch, D., Kirkevåg, A., Kristjansson, J. E., Krol, M., Lauer, A., Lamarque, J. F., Liu, X., Montanaro, V., Myhre, G., Penner, J., Pitari, G., Reddy, S., Seland, Ø., Stier, P., Takemura, T., and Tie, X.: Analysis and quantification of the diversities of aerosol life cycles within AeroCom, Atmos. Chem. Phys., 6, 1777–1813, https://doi.org/10.5194/acp-6-1777-2006, 2006.

• **Section 3.3: The authors should place the calculated values in context with previous measurements. How do these flux values compare to other Arctic or continental regions? A diel cycle of modal flux is an interesting result that needs to be compared to similar and dissimilar calculations.**

We agree with the reviewer that contextualizing the calculated fluxes is important. However, as we have already highlighted in the manuscript, studies reporting aerosol particle fluxes in Arctic or urban snow-covered environments are still very scarce. Flux measurements are inherently site-specific, as they strongly depend on local sources and surface characteristics. This makes direct comparison of absolute flux values across sites problematic.

For this reason, in the literature the more meaningful comparison between sites is often performed using deposition velocity ($V_d$), i.e. the flux normalized by particle concentration, which reduces the influence of site-specific concentration levels and allows a better understanding of how the underlying surface and micrometeorological conditions control deposition. As also noted in previous studies (Mathes et al., 2025), for a general understanding it is more instructive to compare typical deposition velocities over different surface types rather than absolute flux values. In line with this approach, we also compare our calculated deposition velocities with observations reported in the literature, as shown in Sect. 3.4 (Fig. 7), rather than focusing solely on the absolute fluxes.

Mathes, T., Guy, H., Prytherch, J., Kojoj, J., Brooks, I., Murto, S., Zieger, P., Wehner, B., Tjernström, M., and Held, A.: Particle flux–gradient relationships in the high Arctic: emission and deposition patterns across three surface types, Atmos. Chem. Phys., 25, 8455–8474, https://doi.org/10.5194/acp-25-8455-2025, 2025.

**• Lines 336-337: Please state the quantified difference between FACC during AC compared to the other periods that is referenced here.**

In the revised manuscript we now quantify this difference: $F_{ACC}$ during the anticyclonic period is on average 3.4 times higher compared to the cyclonic period, and 1.4 times higher compared to the transition period $T_2$ (Table 4 and Table 5).

**• Figure 3 and 5: Please denote that (if I assume correctly) dots in the box plots represent the mean and line in the box is the median.**

We thank the reviewer for this observation. We confirm that in all box plots, the line inside the box represents the median value, while the dots indicate the mean. We have clarified this in the revised figure captions.

The line inside each box indicates the median, while the squares represent the mean values.

**• Lines 358-361: The authors suggest wind-driven resuspension as a potential mechanism contributing to the variability in FQ-CRS. Did the authors observe more variable FQ-CRS with variable or stronger wind speeds?**

In response to this comment, we selected $F_{Q\text{-}CRS}$ values under conditions of high and low wind speed. The mean fluxes are 0.02 cm$^{-2}$ s$^{-1}$ and $-3.92\times10^{-5}$ cm$^{-2}$ s$^{-1}$, respectively. This indicates that under high wind conditions (> 2.35 m s$^{-1}$, where the mean wind velocity was 2.35 m s$^{-1}$) the flux is more than two orders of magnitude larger and positive (consistent with wind-driven resuspension), whereas under low wind speeds (< 2.35 m s$^{-1}$) it is close to zero and slightly negative (indicating deposition). We have added a brief discussion in the revised manuscript to clarify the relationship between $F_{Q\text{-}CRS}$ variability and wind conditions.

Periods with stronger or more variable winds were associated with increased variability in $F_{Q\text{-}CRS}$. Selecting $F_{Q\text{-}CRS}$ by wind speed, the mean flux was 0.02 cm$^{-2}$ s$^{-1}$ under high wind conditions (> 2.35 m s-1), while it was close to zero and slightly negative ($-3.92\times10^{-5}$ cm$^{-2}$ s$^{-1}$) under low wind speeds (< 2.35 m s-1), supporting the hypothesis of wind-driven resuspension of particles from the surface contributes to the observed fluctuations in deposition fluxes.

**• Lines 360-361: The authors spend little time discussing the differences in the flux diurnal cycle between synoptic regimes. I think the larger fluxes throughout the day during the anticyclonic regime is consistent with the overall finding of higher number concentrations and higher fluxes during the anticyclonic regime and should be stated.**

We agree that the diurnal cycle of fluxes is an important aspect to emphasize. Following the suggestion, we have expanded the discussion by explicitly stating that the larger fluxes observed throughout the day during the anticyclonic regime are consistent with the higher number concentrations and overall higher fluxes already reported for this regime. The following revised text has been added in the new version of the manuscript.

The comparison of the diurnal cycle between the two synoptic regimes further highlights the role of large-scale circulation in controlling particle concentrations and exchange processes. During anticyclonic conditions, fluxes remain consistently higher throughout the day for the ACC particles, and on average higher between 0:00 and 6:00 LT for UFP and Q-CRS particles, during a time of the day when wind speed and TKE are enhanced during the AC period. The larger fluxes observed in the

anticyclonic period with respect to the cyclonic conditions therefore not only reflects the overall increase in particle number concentrations but also suggests more favourable micrometeorological conditions for upward and downward transport, such as enhanced turbulence and stronger coupling between the surface and the boundary layer.

**• Line 384-385: Citation to previous literature should be provided here for this claim.**

Citations to relevant previous studies have been provided later in the manuscript, a few lines below, where the broader context of the study is analysed and compared to existing literature. We chose to introduce these references in that section to maintain a logical flow of discussion.

**• Line 397-400: Are the deposition velocities cited from previous measurements in these lines from Arctic/polar regions? Please clarify.**

Yes, all deposition velocities cited in these lines refer to measurements conducted in remote Arctic or Antarctic locations. We have clarified this in the revised manuscript to make explicit that these values come from polar environments, ensuring a consistent comparison with the present study.

**• Line 401-402: Are the authors able to correlate the parameterized deposition velocities using Slinn (1982) with the calculated median values shown in Fig. 5b from this study? How well do they agree? Further, are the author able to speculate on why the Slinn parameterization has such (visually) great agreement with your calculated values from these measurements?**

To address the comment, we compared our calculated median deposition velocities with the values obtained using the Slinn (1982) parameterization (SL82) at the same particle diameters. The scatter plot (Fig. R1) of observed vs parameterized values shows a very strong agreement, with a Pearson correlation coefficient of 0.96. This indicates that the SL82 parameterization can reproduce our observations with remarkable accuracy.

As for the reason behind this close match, we do not have a definitive explanation. However, we note that the SL82 formulation explicitly accounts for deposition processes relevant under the conditions of our study, such as Brownian diffusion and impaction, which might dominate the observed particle size ranges. This may contribute to the good agreement found here.

[Figure]

Fig. R1. Scatter plot of observed median deposition velocities vs. values parameterized with Slinn (1982, SL82) at the same particle diameters.

Slinn, W.: Predictions for particle deposition to vegetative canopies, Atmos. Environ., 16, 1785–1794, https://doi.org/10.1016/0004-6981(82)90271-2, 1982.

• **Figure 8 caption: it should be clarified that Vn is the deposition velocity normalized by the friction velocity.**

$V_n$ is indeed defined as the deposition velocity normalized by the friction velocity ($V_n = V_d / u^*$). This definition was already provided in the main text (line 425), and we believe it is unnecessary to repeat it in the figure caption to avoid redundancy.

• **Line 435: I might suggest referring to the 0.54-0.89 μm size range as the "large" accumulation mode here.**

We wish to thank the Reviewer for his/her valuable suggestion. Now, in the revised version of the manuscript, we added large to accumulation mode.

• **Line 438: I understand the suggestion that physical characteristics (like different particle densities) may affect the deposition behaviour in this mode, but can the authors clarify why hygroscopicity would affect the deposition behaviour? A citation might be helpful here.**

Hygroscopicity can influence particle deposition because it affects particle size and, consequently, the mechanisms controlling deposition. Under conditions of higher relative humidity, hygroscopic particles can grow due to water uptake, increasing their effective diameter. This growth enhances gravitational settling and impaction processes and may also alter the particle's ability to follow

turbulent eddies, thereby changing deposition efficiency in the accumulation mode. For example, hygroscopic growth has been shown to significantly modify particle size distributions and deposition rates (e.g., Seinfeld and Pandis, 2016). Thus, differences in hygroscopicity between sites or particle types could contribute to the observed differences in deposition behaviour. In the revised version of the manuscript a literature citation has been added.

Seinfeld, J.H., Pandis, S.N.: Atmospheric Chemistry and Physics: From Air Pollution to Climate Change, Wiley, New Jersey, 2016.

• **Lines 438-441: Citations are needed here to allude to these suggested effects.**

We highlight that under polar conditions, electric charging processes induce surface charges on ice and snow particles. Such charging enhances electrostatic attraction of other particles, potentially altering deposition dynamics - especially for intermediate-sized particles that are most sensitive to changes in adhesion and transport mechanisms. In the revised version of the manuscript a literature citation has been added.

Tkachenko, K. and Jacobi, H.-W.: Electrical charging of snow and ice in polar regions and the potential impact on atmospheric chemistry, Environ. Sci.: Atmos., 4, 144, https://doi.org/10.1039/d3ea00084b, 2024

• **Fig 9: The authors should provide units for the y-axis in panel d).**
In panel d), the same y-axis displays both deposition velocity ($V_d$) and sensible heat flux (H), which have different units. Therefore, units were not indicated on the axis itself to avoid confusion. However, the units can be clearly inferred from the main text: $V_d$ is expressed in mm s$^{-1}$, and H in W m$^{-2}$. We have added a clarifying note in the figure caption to make this explicit.

Panel d) shows both deposition velocity ($V_d$, mm s$^{-1}$) and sensible heat flux (H, W m$^{-2}$) on the same y-axis; units for each variable are given here and can also be found in the main text.

• **Fig 10 caption: It should be specified that panels (b,d) are zoomed in versions of panel (a,c).**
Thank you for this suggestion. In the revised version of the manuscript a clarification has been added in the caption of Fig. 10 and Fig. 11

In (b) and (d) a zoom of the graph in (a) and (c) panels is reported.

---

## Author Comment (AC2)

**Article Discussion comments for: "Aerosol dry deposition fluxes on snow during the ALPACA campaign in Fairbanks, Alaska", Donateo et al., Atmospheric Chemistry and Physics, Manuscript ID: egusphere-2025-1366**

We thank the Reviewer for his/her careful assessment of our manuscript and his/her valuable suggestions. We found them very useful for improving and clarifying some of the information that was unclear in the submitted manuscript. Below, we respond to the comments in turn, and summarise modifications made to the manuscript. Our responses are formatted as follows:

Reviewer comments **- black text (bold)**

Author responses - black text

Revised manuscript text - green text

Line numbers refer to those in the original submission.

Reviewer #2: RC2: "Comment on egusphere-2025-1366' - https://doi.org/10.5194/egusphere-2025-1366-RC2

**This study looked at how aerosols exchange near the surface in a cold, snowy Arctic city, Fairbanks, Alaska. The authors used EC during a 23-day period in winter 2022 to measure whether particles were rising into the air (emitted) or falling back to the ground (deposited). They studied different particle sizes, including ultrafine particles, mid-sized particles, and larger ones that are more likely to settle.**

**They found that the smallest particles were mostly emitted, especially during the day, likely due to traffic and other local sources. They also found that the bigger particles behaved differently. They were often depositing to the ground, especially when the weather was calm and cold (AC). But these upward and downward movements didn't always match up with the particle concentration in the air, showing that just looking at concentration isn't enough to understand pollution. Importantly, they discovered that synoptic conditions played a major role. When the wind was weak and the air was still, pollution built up near the ground. But stronger winds and temperature differences between the air and the snowy ground helped push particles back down, even if there weren't many in the air.**

**A major finding was that downward movement (deposition) didn't just happen because of gravity. Instead, turbulence in the air was a key driver. In extremely cold conditions, sharp temperature differences near the snowy surface may have created an extra push for particles to settle, a process known as thermophoresis. This may be a missing piece in current models, potentially leading to an underestimation of dry deposition rates.**

**Overall, the manuscript is well structured and scientifically sound. I have some comments that I believe, once addressed, will further strengthen the clarity and impact of the study and bring it to a publishable standard.**

**Major comment:**

**The manuscript discusses the likely influence of local sources (e.g., traffic, UAF activity) and regional sources (e.g., downtown Fairbanks) based primarily on the diurnal patterns. While these observations are suggestive, they remain qualitative. To strengthen the attribution of aerosol sources under different synoptic regimes, I suggest incorporating back trajectory analyses (e.g., using HYSPLIT or similar tools). Stratifying trajectories by synoptic condition and/or flux event (e.g., high UFP emissions or deposition episodes) would provide more robust evidence for the origin of aerosols and help distinguish between local vs. advected pollution events. This would also complement the existing discussion of transport under anticyclonic vs cyclonic regimes. If high-resolution meteorological data to run the back trajectory analysis is not available to resolve local-scale features, extending the footprint-based approach separately under different synoptic conditions would still provide a valuable, site-relevant alternative. A brief discussion of typical source characteristics by particle size (e.g., UFP, ACC, and Q-CRS) would also be helpful to support this analysis.**

We agree that back trajectory analyses can provide useful insights into the origin of aerosols. However, Brett et al., (2025) already showed that WRF/Chem frame can trace the distribution of pollution sources in Fairbanks during ALPACA campaign to a large extent. HYSPLIT therefore becomes not much informative. To investigate the origin of the air masses during the study period, we performed a pollution rose analysis (a footprint-based approach), which can turn useful to support a brief discussion about the sources from downtown vs from outside the Fairbanks basin. Further, we include in the Appendix more concentration trends for UFP, ACC and Q-CRS overlapped with the time trends of tracers, such as BTEX as tracers of traffic emissions. Clearly, UFP shows a good association with the BTEX, especially toluene, which is the best traffic tracer, while Q-CRS not that

much. ACC and Q-CRS better overlap with benzene which is also affected by biomass burning and possibly by a regional background. This analysis confirms that the observed patterns are largely driven by local sources rather than long-range transport. While long-range advection cannot be fully excluded, our results together with the synoptic and local-scale meteorological context indicate that the contribution of regional transport is minor compared to local emissions. In the revised version of the manuscript these aspects are now discussed.

To investigate source contributions in greater detail, we applied a footprint-based analysis. Specifically, we used bivariate polar plots (pollution roses; Fig. A1) to examine the dependence of UFP, ACC, and Q-CRS particle number concentrations on wind speed and wind direction at the UAF Farm site, stratified by synoptic regime. This approach provides a clear visualization of how different meteorological conditions modulate source impacts. Under anticyclonic conditions, the pollution roses consistently point to downtown Fairbanks as the dominant source region for all three particle size classes. The highest concentrations occurred at low wind speeds, particularly for air masses arriving from the S–SE sector, indicative of stagnant conditions that favour the accumulation of locally emitted particles. An additional component is associated with air masses transported from the W–NW sector under stronger winds. These flow conditions are linked to rural areas outside the Fairbanks basin or to traffic sources in the Goldstream (a tributary valley of the Tanana basin). This source apportionment is consistent with the behaviour of BTEX compounds (benzene, toluene, ethylbenzene and xylene) measured during the ALPACA campaign. In particular, UFP concentrations showed a strong association with toluene (Fig. A2a), the most reliable traffic tracer, especially under anticyclonic conditions when calm winds from the S–SE favoured the accumulation of urban emissions from downtown Fairbanks. By contrast, Q-CRS particles exhibited only a weak relationship with BTEX (not shown here), reflecting the contribution of sources other than local traffic. ACC and Q-CRS particles displayed (Fig A2b,c) a better correspondence with benzene, which in the Fairbanks area is influenced not only by traffic but also by biomass burning and regional background transport. Episodes of enhanced ACC and Q-CRS concentrations not mirrored by BTEX further support the presence of additional, non-traffic sources affecting these particle size classes.

[Figure]

**Fig. A1.** Bivariate polar plots (pollution roses) relating (a) UFP, (b) ACC and (c) Q-CRS particles mode in the anticyclonic (upper panels) and cyclonic (bottom panels) period.

[Figure]

**Fig. A2. Time series of the number particle concentration for (a) UFP, (b) ACC and (c) Q-CRS size range. In (a) on the right axis the toluene concentration (in ppb) is reported. In (b) and (c) is reported, on the right axis, the benzene concentration (in ppb)**

**Throughout the manuscript, inversion layers and stability regimes and vertical profile of atmosphere are discussed as key factors influencing aerosol fluxes. However, the meteorological context is not sufficiently demonstrated. I suggest including synoptic-scale information, such as MSLP composites, and vertical profiles (e.g., radiosonde soundings), to better illustrate the evolution and structure of inversion layers during the campaign periods for different dominant synoptic conditions.**

We appreciate the reviewer's suggestion to include synoptic-scale meteorological information. In Sect. 3.1, we have already provided a detailed description of the synoptic conditions during the campaign periods, with explicit references to previous studies that extensively analysed the large-scale meteorological context and its relation to inversion layer formation. To avoid redundancy, we did not repeat the full synoptic analysis in the present manuscript, but instead we relied on those established results to frame the interpretation of our flux data. In addition, the composite radiosonde profiles presented in Brett et al. 2025 (see their Fig. 3) directly illustrate the occurrence and structure of inversion layers during our campaign periods, and we refer the reader to that analysis for a

comprehensive depiction. We believe this provides sufficient meteorological context for the scope of the paper, while keeping the focus on aerosol fluxes and deposition.

Brett, N., Law, K. S., Arnold, S. R., Fochesatto, G. J., Raut, J.-C., Onishi, T., Gilliam, R., Fahey, K. Huff, D., Pouliot, G., Barret, B., Dieudonné, E., Pohorsky, R., Schmale, J., Baccarini, A., Bekki, S., Pappaccogli, G., Scoto, F., Decesari, S., Donateo, A., Cesler-Maloney, M., Simpson, W., D'Anna, B., Temime-Roussel, B., Savarino, J., Albertin, S., Mao, J., DeCarlo, P. F., Selimovic, V., and Yokelson, R.: Investigating processes influencing simulation of local Arctic wintertime anthropogenic pollution in Fairbanks, Alaska, during ALPACA-2022, Atmos. Chem. Phys., 25, 1063–1104, https://doi.org/10.5194/acp-25-1063-2025, 2025.

**Minor comments:**

**The manuscript does not discuss whether EC has previously been used in snow-covered or Arctic urban environments, nor does it acknowledge any potential limitations of using this technique under such conditions. If this study is among the first to apply EC to aerosol deposition in Arctic snow-covered urban areas, it would be helpful to state this clearly and reflect on associated uncertainties.**

In the Introduction (Sect. 1), we have already emphasized the current state of the art and highlighted that only very few studies have applied the eddy covariance technique to aerosol fluxes over snow-covered and Arctic (urban) environments. Our work therefore contributes to filling this gap. Nevertheless, following the Reviewer's suggestion, we have revised the text to make this aspect more explicit, and we have added a short discussion of the potential limitations and uncertainties related to the application of EC under such conditions.

This makes our work one of the few attempts to investigate deposition processes under such conditions … At the same time, it is important to acknowledge that EC measurements in these environments may be subject to additional uncertainties, for example due to low turbulence, surface heterogeneity, or snow-related effects on particle exchange. These aspects are discussed in the manuscript to clarify both the novelty and the limitations of the present study.

**Figure 1 shows the spatial distribution of the flux footprint, but it lacks clarification of what the coloured contours represent.**

We have revised the figure caption to clarify that the coloured contours in Fig. 1 represent the relative contribution of each area to the measured flux at the tower, with warmer colours indicating higher contribution to the flux footprint.

Coloured contours in the footprint image represent the relative contribution of each area to the measured flux at the tower, with warmer colours indicating higher contribution.

**In Section 2.1, you mention several nearby potential aerosol sources, including the Tanana River. To support later discussions on aerosol sources under different synoptic conditions, I suggest highlighting these key features (e.g., the river, airport, major roads) directly to Figure 1. This would provide helpful spatial context when interpreting the footprint and probably back trajectories influences on observed fluxes.**

While we agree that highlighting specific sources would provide additional context, the zoom level and spatial extent chosen for Figure 1 do not include some of these features, such as power plants and the airport, which are located outside the mapped area. Including them would therefore not be feasible without changing the scale of the figure. The main text (Sect. 2.1) provides a detailed description of nearby potential sources, which, together with the footprint, offers sufficient spatial context for interpreting the fluxes under different synoptic conditions. However, a map at this finer scale can be found in Brett et al. (2025, Fig. 1), and we have added a sentence in the text of the revised manuscript to refer the reader to this source for additional spatial detail.

A more detailed map of local sources, including the airport and power plants, can be found in Brett et al. (2025, Fig. 1).

Brett, N., Law, K. S., Arnold, S. R., Fochesatto, G. J., Raut, J.-C., Onishi, T., Gilliam, R., Fahey, K. Huff, D., Pouliot, G., Barret, B., Dieudonné, E., Pohorsky, R., Schmale, J., Baccarini, A., Bekki, S., Pappaccogli, G., Scoto, F., Decesari, S., Donateo, A., Cesler-Maloney, M., Simpson, W., D'Anna, B., Temime-Roussel, B., Savarino, J., Albertin, S., Mao, J., DeCarlo, P. F., Selimovic, V., and Yokelson, R.: Investigating processes influencing simulation of local Arctic wintertime anthropogenic pollution in Fairbanks, Alaska, during ALPACA-2022, Atmos. Chem. Phys., 25, 1063–1104, https://doi.org/10.5194/acp-25-1063-2025, 2025.

**Please clarify whether a constant $z = 11$ m (instrument height) was used for the calculation of the atmospheric stability parameter $\zeta = z/L$. Also, briefly describe how the Obukhov length L was derived.**

Indeed, the instrument height (z = 11 m) was used as a constant for the calculation of the atmospheric stability parameter $\zeta$ = z / L. The Obukhov length L was derived following the Monin–Obukhov similarity theory, using the standard formulation:

$$L = \frac{-u_*^3 \overline{T}}{\kappa g \overline{w'T_s'}}$$

where u* is the friction velocity, $\overline{w'T_s'}$ is the kinematic virtual potential temperature flux, $\overline{T}$ the mean virtual potential temperature, $\kappa$ the von Kármán constant, and g the gravitational acceleration. We have added a reference in Sect. 2.3 to clarify the calculation of the Monin Obukhov length (L).

Stull, R. B.: An introduction to boundary layer meteorology, Kluwer Academic Publishers, Dordrecht, 1988.

**Regarding the data processing and all filtration processes described in lines 171 to 202, clarify on how representative the final dataset is. Especially whether there is sufficient coverage across stability regimes, times of day, or meteorological conditions. Also in line 171, you mention that data were discarded for being "outside the absolute limits." Could you clarify what specific thresholds were used in this step?**

The percentage of data available for the analysis, expressed relative to the total dataset, has already been specified in the manuscript, in Sect. 2.3. The final dataset retains a significant fraction of the measurements, which is consistent with similar experiments in polar environments, where extreme conditions often lead to unavoidable data losses. In the revised manuscript (caption in Fig. 5) we now clarify the representativeness of the dataset. The percentage of valid data is 67% during the anticyclonic period and 74% during the cyclonic period, ensuring sufficient coverage for a robust comparison between regimes. On average, the coverage of valid data for each hour of the day is 67% (AC) and 74% (C) for UFPs, 65% (AC) and 84% (C) for ACC, and 64% (AC) and 81% (C) for Q-CRS. These values confirm that sufficient coverage is available across diurnal cycles and synoptic regimes to support our analysis.

Regarding data discarded "outside the absolute limits," the thresholds applied were based on instrument specifications and physical plausibility criteria: for example, particle concentrations below the detection limit or exceeding the maximum measurable range, and extreme flux or micrometeorological values inconsistent with the expected measurement range.

**Please clarify on the average error of 101% in line 193.**

The reported "average error of 101%" refers to the relative uncertainty of the specific measurement, rather than an absolute error. This high percentage reflects the low counting typical for quasi coarse particle fluxes under the extreme polar conditions encountered and is consistent with previous studies in similar environments (Donateo et al, 2023).

Donateo, A., Pappaccogli, G., Famulari, D., Mazzola, M., Scoto, F., and Decesari, S.: Characterization of size-segregated particles turbulent flux and deposition velocity by eddy correlation method at an Arctic site, Atmos. Chem. Phys., 23, 7425–7445, https://doi.org/10.5194/acp-23-7425-2023, 2023.

**The final sentence on page 16 states that particle fluxes during the cyclonic period are "on average null for the whole day" (referring to Fig. 6). However, this statement does not appear to be consistent with panel (a) of Fig. 6 - for UFPs. Please clarify or revise this statement for accuracy.**

We agree that qualitative expressions such as "on average null" can be misleading and that numerical values improve clarity. We calculated the hourly mean value over the diurnal cycle of the net particle flux in the three size classes for AC and C. We obtain that during the cyclonic period particle fluxes in all three size modes result lower, especially for the ACC and Q-CRS modes, representing 34% and 28% of the corresponding ones in AC regimes, respectively. Under cyclonic regimes the UFP flux diurnal pattern decreased on average to 65% of the anticyclonic one. In the revised manuscript, we have modified the sentence to make this point clearer.

On the other hand, during the cyclonic period particle fluxes in all three size modes result lower, especially for the ACC and Q-CRS modes, representing 34% and 28% of the corresponding ones in AC regimes, respectively (Fig. 6). Net particle fluxes remained consistently positive on average, for UFPs during the cyclonic period.

**In the conclusion of Section 3.2, it has been stated that the UAF Farm site is devoid of local sources and that aerosol concentrations are mainly influenced by transport. However, earlier in the manuscript (e.g., Section 2.1), the potential influence of local sources such as the nearby airport has been discussed. Please clarify what is meant by "local" in this context and reconcile these two statements to avoid confusion.**

In Sect. 3.2, the term "devoid of local sources" was intended in relative terms with respect to the "neighbouring residential districts" located downtown. Our measurements indicate that local sources in UAF - as defined by the EC footprint in Fig. 1 (roughly 1 km wide) - in fact exist and account for the emissive particle fluxes, especially in the UFP size range. Nevertheless, such sources represent a small percentage of the total combustion emissions (e.g. traffic, residential and power plants) in the Fairbanks basin (around 20 km wide) as discussed in Brett et al. (2025) and indicated by the little association between surface inversions and PM2.5 concentrations in UAF (Simpson et al., 2024). We have revised the text to clarify this distinction, specifying that the site is largely free of strong direct emission sources with respect to the total emissions in the Fairbanks area, while acknowledging that minor anthropogenic or natural sources exist in the surroundings.

This can be partly explained by the limited presence of significant direct emission sources within the source area footprint of the UAF Farm site with respect to the total emissions in the Fairbanks basin. Aerosol concentrations are influenced by the transport from the neighbouring residential districts (with heating and transportation) and/or from the power plants in the extended area.

Simpson, W. R., Mao, J., Fochesatto, G. J., Law, K. S., DeCarlo, P. F., Schmale, J., Pratt, K. A., Arnold, S. R., Stutz, J., Dibb, J. E., Creamean, J. M., Weber, R. J., Williams, B. J., Alexander, B., Hu, L., Yokelson, R. J., Shiraiwa, M., Decesari, S., Anastasio, C., D'Anna, B., Gilliam, R. C., Nenes, A., St. Clair, J. M., Trost, B., Flynn, J. H., Savarino, J., Conner, L. D., Kettle, N., Heeringa, K. M., Albertin, S., Baccarini, A., Barret, B., Battaglia, M. A., Bekki, S., Brado, T. J., Brett, N., Brus, D., Campbell, J. R., Cesler-Maloney, M., Cooperdock, S., Cysneiros de Carvalho, K., Delbarre, H., DeMott, P. J., Dennehy, C. J. S., Dieudonné, E., Dingilian, K. K., Donateo, A., Doulgeris, K. M., Edwards, K. C., Fahey, K., Fang, T., Guo, F., Heinlein, L. M. D., Holen, A. L., Huff, D., Ijaz, A., Johnson, S., Kapur, S., Ketcherside, D. T., Levin, E., Lill, E., Moon, A. R., Onishi, T., Pappaccogli, G., Perkins, R., Pohorsky, R., Raut, J.-C., Ravetta, F., Roberts, T., Robinson, E. S., Scoto, F., Selimovic, V., Sunday, M. O., Temime-Roussel, B., Tian, X., Wu, J., and Yang, Y.: Overview of the Alaskan Layered Pollution and Chemical Analysis (ALPACA) Field Experiment, ACS EST Air, 1, 200–222, https://doi.org/10.1021/acsestair.3c00076, 2024.

**In lines 306–314, the elevated night time accumulation-mode particle fluxes are attributed to re-entrainment under stable, calm conditions. While this is a plausible mechanism, the role of varying emissions, such as increased night time residential heating or episodic sources is not discussed. How did you ignore considering whether higher night time emissions may also contribute to the observed fluxes, particularly since these could be source-driven rather than solely due to surface-layer turbulence.**

We would like to clarify that the role of emissions was in fact already acknowledged in the original manuscript. Specifically, in lines 316–317 (Sect. 3.2, original submitted paper) we noted that *"A secondary maximum observed for $N_{ACC}$ particles in the evening hours can be due specifically to residential heating sources"*. In addition, in lines 353–355 (Sect. 3.3, original submitted paper) we wrote that *"A secondary maximum was observed for FUFP and FACC in the evening hours (around 18:00 LT) that can be due also to the evening rush hour (Fig. 6a and b). In general, the presence of a correspondent peak in the FUFP and FACC mode particles indicates that a common emissive source was present in the area in those days"*.

To make this aspect clearer, we have revised the text to better emphasize that nighttime and evening flux enhancements can indeed reflect not only surface-layer turbulence and re-entrainment but also increased emissions. In particular, residential heating in Fairbanks has a well-documented diurnal cycle that follows occupancy patterns of domestic and office spaces (Ketcherside et al., 2025). We have therefore updated the relevant sentences and added this reference to explicitly highlight the contribution of emission-driven variability.

Ketcherside, D. T., Yokelson, R. J., Selimovic, V., Robinson, E. S., Cesler-Maloney, M., Holen, A. L., Wu, J., Temime-Roussel, B., Ijaz, A., Kuhn, J., Moon, A., Pappaccogli, G., Cysneiros de Carvalho, K., Decesari, S., Alexander, B., Williams, B. J., D'Anna, B., Stutz, J., Pratt, K. A., DeCarlo, P. F., Mao, J., Simpson, W. R., Hopke, P. K., Hu, L.: Wintertime abundance and sources of key trace gas and particle species in Fairbanks, Alaska, J. Geoph. Res.: Atmos., 130, e2025JD043677, https://doi.org/10.1029/2025JD043677, 2025

**Since the mean values are available in the boxplots in Figure 3 & 5 (I assume the dots are the mean values), Table 1 may no longer be necessary (maybe keeping it in the supplementary materials). Presenting this information visually would improve clarity and reduce redundancy.**

We agree that the mean values are shown in the boxplots; however, Table 1 reports additional statistical information, such as standard deviation, standard error, and percentile ranges, which cannot be directly inferred from the figures. Therefore, we prefer to keep Table 1 in the main text to provide a more comprehensive summary of the dataset. We have clarified in the revised figure captions that the squares (or dots) represent the mean values.

The line inside each box indicates the median, while the squares represent the mean values.

**In Figure 6, the diurnal variability in emission fluxes is interpreted primarily as a result of traffic emissions, based on the timing of observed peaks. While this is plausible, I recommend**

**also considering the role of deposition (i.e., net negative fluxes) and turbulence strength (e.g., vertical wind speed or shear) in shaping the diurnal pattern. Do they show similar consistency? Including these factors would provide a more complete picture of the boundary-layer processes controlling flux variability, and help to better separate source-driven vs. transport/mixing influences.**

We agree that deposition and turbulence strength are important factors influencing the diurnal cycle of particle fluxes. In the revised version of Fig. 6, we now report the net particle fluxes (including both positive and negative values) instead of only emission fluxes. In addition, we have included the diurnal pattern of vertical turbulence intensity ($\sigma_w$), which reflects the role of shear-driven turbulence. This integrated analysis shows that flux peaks often coincide with enhanced turbulence, indicating that surface-layer mixing contributes to the observed diurnal variability alongside traffic-related emissions and deposition. We have revised the manuscript to include these patterns (Sect. 3.3, Fig. 6) and discuss the combined influence of emissions, deposition, and turbulence on the diurnal flux cycle.

To better disentangle the role of sources and boundary-layer processes, we included the diurnal pattern of vertical turbulence intensity ($\sigma_w$) in Fig. 6. The results show that peaks in fluxes often coincide with enhanced turbulence, indicating that surface-layer mixing contributes to the observed diurnal variability alongside traffic emissions and deposition. These observations help separate source-driven signals from processes related to deposition and turbulent transport (Fig. 6).

[Figure]

**Figure 6: Diurnal cycles (mean) of particle flux (net values, including both positive and negative fluxes; left axis) and standard deviation of vertical wind speed ($\sigma_w$; right axis for each panel) at UAF-Farm for a) $F_{UFP}$, b) $F_{ACC}$ and c) $F_{Q-CRS}$. The diurnal trends are shown both for the anticyclonic and cyclonic period. Continuous lines are the mean value. The shadow area represents the standard error. Hour is in standard local time (LT).**

**In Section 3.4 and Table 2, the transfer velocity v=F/N is only presented as a mean value. For consistency with Figures 3 and 5, and to better capture variability and uncertainty, why not present them as a boxplot grouped by stability class and particle size. This would allow readers**

**to more directly compare patterns in v with those in N and F, and assess whether the observed means are representative of the distribution.**

To provide a more comprehensive view of variability and uncertainty, we have prepared boxplots of the deposition velocity ($V_d$) and normalized deposition velocity ($V_n$), grouped by mesoscale meteorology classes (AC, C, T) and particle size (UFP, ACC, Q-CRS). These figures are included in the Appendix A (Fig. A4 and Fig. A5) to complement Table 2. This allows readers to directly compare the distributions of $V_d$ and $V_n$ with the corresponding patterns in particle number concentrations (N) and fluxes (F), and to evaluate whether the mean values reported in Table 2 are representative of the observed variability. The figures included in the Appendix have now been explicitly referenced in the main text.

[Figure]

**Figure A4: Daily box plots for (a) Vd$_{UFP}$, (b) Vd$_{ACC}$ and (c) Vd$_{Q-CRS}$. Boxes represent the 25th and 75th percentiles. Whiskers correspond to ± 2.7σ and 99.3% data coverage. The colour bands indicate the anticyclonic period AC (red), the cyclonic period C (blue) and the two transition periods T (green). The line inside each box indicates the median.**

[Figure]

**Figure A5: Daily box plots for (a) Vn$_{UFP}$, (b) Vn$_{ACC}$ and (c) Vn$_{Q-CRS}$. Boxes represent the 25th and 75th percentiles. Whiskers correspond to ± 2.7σ and 99.3% data coverage. The colour bands indicate the anticyclonic period AC (red), the cyclonic period C (blue) and the two transition periods T (green). The line inside each box indicates the median.**

**Why haven't you had included time series plots of V$_n$, Vd, and friction velocity (u*)? This would help the reader assess how these variables evolve and interact dynamically, especially under changing meteorological conditions.**

While we have not included full time series of V$_n$, V$_d$, and u* in the main text, we provide boxplots of V$_d$ (Fig. A4), V$_n$ (Fig. A5), u* (Fig. A6a) and the turbulent kinetic energy TKE (Fig. A6b) grouped by mesoscale meteorology and particle size in the Appendix A. These visualizations summarize the variability and allow readers to assess the typical range and distribution of the deposition velocity under different atmospheric conditions, providing insight into their interaction with turbulence. Given the large dataset and high temporal resolution, we believe that these boxplots offer a clear and concise representation of the dynamic behaviour without overloading the manuscript with extensive time series plots.

[Figure]

**Figure A6: Daily box plots for (a) friction velocity u*, and (b) turbulent kinetic energy TKE. Boxes represent the 25th and 75th percentiles. Whiskers correspond to ± 2.7σ and 99.3% data coverage. The colour bands indicate the anticyclonic period AC (red), the cyclonic period C (blue) and the two transition periods T (green). The line inside each box indicates the median, while the dots represent the mean values.**

**Where is equation 3 mentioned in line 429?**

We thank the reviewer for noticing this. The reference to Equation 3 in line 429 was a remnant from previous manuscript versions and has been removed in the revised manuscript.

**In Section 3.6, the discussion of meteorological and deposition differences across regimes (e.g., ACa, ACb, C, T1, T2) is presented primarily in qualitative terms (e.g., "median u* was higher...", "Vd values were greater…"). I suggest making this section more quantitatively grounded by explicitly referencing values from Tables 4 and 5 within the text to support these comparisons.**

In the revised manuscript, we have added explicit numerical values from Tables 4 and 5 to support the discussion of differences in meteorological conditions and deposition velocities across the stability regimes. This makes the comparisons more quantitative and allows readers to directly assess the magnitude of differences in friction velocity, $V_d$, and particle fluxes between regimes.

**Additionally, to assess whether the observed differences are statistically meaningful, I recommend performing basic statistical tests on key variables such as deposition velocity or ΔT. This would help substantiate the conclusions about the influence of meteorological regimes on aerosol deposition and strengthen the interpretation of the results.**

To quantify the statistical significance of differences across meteorological regimes, we have performed non-parametric tests (Kruskal–Wallis followed by post-hoc pairwise comparisons) on key variables, including friction velocity ($u^*$), temperature inversion ($\Delta T$), particle fluxes (F) and deposition velocity ($V_d$). The results indicate that several differences among regimes are statistically significant ($p < 0.05$), even if $V_d$s did not differ significantly among meteorological regimes ($p > 0.05$). These results have been added to the revised manuscript in Sect. 3.6.

**The observation that median deposition velocities for ACC particles are lower than for UFPs is counterintuitive given their larger size. This might reflect differences in turbulence coupling, rebound effects, or diffusional deposition mechanisms, but it would be helpful if the authors could provide a brief discussion or hypothesis to interpret this behaviour.**

We agree that the lower median deposition velocities observed for accumulation mode (ACC) particles compared to ultrafine particles (UFPs) may seem counterintuitive, since gravitational settling typically increases with particle size. The lower median deposition velocities observed for accumulation mode particles compared to ultrafine particles may result from differences in deposition mechanisms. UFPs (< 100 nm) are more strongly affected by Brownian diffusion, which can enhance deposition rates to surfaces, particularly under stable conditions with low turbulence. In contrast, ACC particles (~250–800 nm) exhibit weaker diffusional transport and are more reliant on turbulent impaction and interception, processes that may be less efficient under the relatively low-friction velocity conditions observed during much of the campaign. Additionally, partial rebound or resuspension of ACC particles upon surface contact cannot be excluded, especially for snow, potentially lowering their net measured deposition velocity. These combined effects can explain the reversal of the expected size-dependence in median deposition velocities, as also shown from many existing parameterization (see Fig. 7b)

**While the surface-based eddy covariance measurements provide valuable insight into particle emission and deposition processes, the study would benefit from acknowledging the limitation posed by the lack of aerosol flux measurements at the top of the boundary layer. Without data on vertical flux divergence, it is difficult to fully constrain the aerosol budget or to distinguish surface-driven processes from entrainment or dilution effects. A brief discussion of this limitation, and its implications for interpreting the observed surface fluxes, would improve the overall clarity and transparency of the analysis.**

We agree that the absence of aerosol flux measurements at the top of the boundary layer limits the possibility to directly assess vertical flux divergence and to fully separate surface-driven processes from entrainment or dilution effects. At the same time, the PBL structure in this study cannot accurately be described as a mixing layer with entrainment on top of it, as a large variability was observed in the thermal and dynamical structure of the atmosphere in the lower tenths of meters above the ground (Fig. 10). We believe that our study was successful in demonstrating how turbulent mixing and aerosol fluxes can develop (and in which atmospheric layer) under a very stable winter polar atmosphere, although, we acknowledge, measuring the aerosol fluxes at a fixed elevation is not enough to fully constrain depositions. Following the Referee's suggestion, we have added a sentence in the Conclusion (Sect. 4), explicitly acknowledging this limitation and its potential implications for the interpretation of our results.

While our results provide robust evidence of the influence of atmospheric stability on aerosol fluxes, the lack of flux measurements at the top of the mixing layer limits our ability to fully constrain the vertical particle budget. This limitation should be considered when interpreting the relative contributions of surface exchange versus entrainment, divergence or dilution processes, particularly under well-mixed conditions.